# Contextual Games:
# Multi-Agent Learning with Side Information

**Pier Giuseppe Sessa**
ETH Zürich
sessap@ethz.ch

**Ilija Bogunovic**
ETH Zürich
ilijab@ethz.ch

**Andreas Krause**
ETH Zürich
krausea@ethz.ch

**Maryam Kamgarpour**
ETH Zürich
maryamk@ethz.ch

## Abstract

We formulate the novel class of *contextual games*, a type of repeated games driven by contextual information at each round. By means of *kernel-based* regularity assumptions, we model the correlation between different contexts and game outcomes and propose a novel online (meta) algorithm that exploits such correlations to minimize the *contextual regret* of individual players. We define game-theoretic notions of *contextual Coarse Correlated Equilibria* (c-CCE) and *optimal contextual welfare* for this new class of games and show that c-CCEs and optimal welfare can be approached whenever players' contextual regrets vanish. Finally, we empirically validate our results in a traffic routing experiment, where our algorithm leads to better performance and higher welfare compared to baselines that do not exploit the available contextual information or the correlations present in the game.

## 1 Introduction

Several important real-world problems, ranging from economics, engineering, and computer science involve multiple interactions of self-interested agents with coupled objectives. They can be modeled as *repeated games* and have received recent attention due to their connection with learning (e.g., [11]).

An important line of research has focused, on the one hand, on characterizing game-theoretic equilibria and their efficiency and, on the other hand, on deriving fast learning algorithms that converge to equilibria and efficient outcomes. Most of these results, however, are based on the assumption that the players always face the exact same game, repeated over time. While this leads to strong theoretical guarantees, it is often unrealistic in practical scenarios: In routing games [30], for instance, the agents' travel times and hence the 'rules' of the game are governed by many time-changing factors such as network's capacities, weather conditions, etc. Often, players can observe such factors, and hence could take better decisions depending on the circumstances.

Motivated by these considerations, we introduce the new class of *contextual games*. Contextual games define a more general class of repeated games described by different contextual, or side, information at each round, denoted also as *contexts* in analogy with the bandit optimization literature (e.g., [24]). Importantly, in contextual games players can observe the current context before playing an action, which allows them to achieve better performance, and converge to stronger notions of equilibria and efficiency than in standard repeated games.

**Related work.** Learning in repeated *static* games has been extensively studied in the literature. The seminal works [17, 18] show that simple *no-regret* strategies for the players converge to the set of Coarse Correlated Equilibria (CCEs), while the *efficiency* of such equilibria and learning dynamics has been studied in [6, 31]. Exploiting the static game structure, moreover, [38, 15] propose faster learning algorithms, and a long array of works (e.g., [35, 7, 4]) study convergence to Nash equilibria. Learning in *time-varying* games, instead, has been recently considered [14], where the authors show that dynamic regret minimization allows players to track the sequence of Nash equilibria, provided that the stage games are monotone and slowly-varying. Adversarially changing zero-sum games have

also been studied [10], with convergence guarantees to the Nash equilibrium of the time-averaged game. Our contextual games model is fundamentally different than [14, 10] in that we assume players observe the current context (and hence have prior information on the game) before playing. This leads to new equilibria and a different performance benchmark, denoted as *contextual regret*, described by the best *policies* mapping contexts to actions. Perhaps closer to ours is the setup of stochastic (or Markov) games [34], at the core of multi-agent reinforcement learning (see [9] for an overview). There, players observe the *state* of the game before playing but, differently from our setup, the evolution of the state depends on the actions chosen at each round. This leads to a nested game structure, which requires significant computational power and players' coordination to compute equilibrium strategies via backward induction [16, 13]. Instead, we consider arbitrary contexts' sequences (potentially chosen by an adversarial Nature) and show that efficient algorithms converge to our equilibria in a decentralized fashion.

From a single player's perspective, a contextual game can be reduced to a special *adversarial* contextual bandit problem [8, Chapter 4], for which several no-regret algorithms exist. All such algorithms, however, rely on high-variance estimates for the rewards of non-played actions and thus their performance degrades with the number of actions available. A fact not exploited by these algorithms is that in a contextual game *similar contexts and game outcomes likely produce similar rewards* (e.g., in a routing game, similar network capacities and occupancy profiles lead to similar travel times). We encode this fact using kernel-based regularity assumptions and (similarly to [32] in non-contextual games) show that exploiting these assumptions, and additionally observing the past opponents' actions, players can achieve substantially improved performance compared to using standard bandit algorithms. For instance, for $K$ actions and adversarially chosen contexts from a finite set $\mathcal{Z}$, the bandit $\mathcal{S}$-ExP3[8] incurs $\mathcal{O}(\sqrt{TK|\mathcal{Z}|\log K})$ contextual regret, while our approach leads to a $\mathcal{O}(\sqrt{T|\mathcal{Z}|\log K} + \gamma_T\sqrt{T})$ guarantee, where $\gamma_T$ is a sample-complexity parameter describing the degrees of freedom in the player's reward function. For commonly used kernels, this results in a sublinear regret bound that grows only *logarithmically* in $K$. Moreover, when contexts are stochastic and private to a player, we obtain a $\mathcal{O}(\sqrt{T\log K} + \gamma_T\sqrt{T})$ pseudo-regret bound. This bound should be compared to the $\mathcal{O}(\sqrt{TK\log K})$ pseudo-regret of [5] which –unlike us– assumes observing the rewards for non-revealed contexts, and the $\mathcal{O}(\sqrt{cTK\log K})$ pseudo-regret of [28], which assumes known contexts distribution and a linear dependence of rewards on contexts in $\mathbb{R}^c$.

**Contributions.** We formulate the novel class of *contextual games*, a type of repeated games characterized by (potentially) different contextual information available at each round.

- We identify the *contextual regret* as a natural benchmark for players' individual performance, and propose novel online algorithms to play contextual games with no-regret. Unlike existing contextual bandit algorithms, our algorithms exploit the correlation between different game outcomes, modeled via kernel-based regularity assumptions, yielding improved performance.
- We characterize equilibria and efficiency of contextual games, defining the new notions of *contextual Coarse Correlated Equilibria (c-CCE)* and *optimal contextual welfare*. We show that c-CCEs and contextual welfare can be approached in a decentralized fashion whenever players minimize their contextual regrets, thus recovering important game-theoretic results for our larger class of games.
- We demonstrate our results in a repeated traffic routing application. Our algorithms effectively use the available contextual information (network capacities) to minimize agents' travel times and converge to more efficient outcomes compared to other baselines that do not exploit the observed contexts and/or the correlations present in the game.

## 2 Problem Setup

We consider repeated interactions among $N$ agents, or players. At every round, each player selects an action and receives a payoff that depends on the actions chosen by all the players as well as the *context* of the game at that round. More formally, we let $\mathcal{Z}$ represent the (potentially infinite) set of possible contexts, and $\mathcal{A}^i$ be the set of actions available to player $i$. Then, we define $r^i : \mathcal{A} \times \mathcal{Z} \to [0, 1]$ to be the reward function of each player $i$, where $\mathcal{A} := \mathcal{A}^1 \times \cdots \times \mathcal{A}^N$ is the joint action space. Importantly, we assume $r^i$ is *unknown* to player $i$. With the introduced notation, a repeated *contextual game* proceeds as follows. At every round $t$:

- Nature reveals context $z_t$
- Players observe $z_t$ and, based on it, each player $i$ selects action $a_t^i \in \mathcal{A}^i$, for $i = 1, \ldots N$.
- Players obtain rewards $r^i(a_t^i, a_t^{-i}, z_t)$, $i = 1, \ldots, N$.

Moreover, as specified later, player $i$ receives feedback information at the end of each round that it can use to improve its strategy. Let $\Pi^i$ be the set of all policies $\pi : \mathcal{Z} \to \mathcal{A}^i$, mapping contexts to actions. After $T$ game rounds, the performance of player $i$ is measured by the *contextual regret*:

$$R_c^i(T) = \max_{\pi \in \Pi^i} \sum_{t=1}^{T} r^i\big(\pi(z_t), a_t^{-i}, z_t\big) - \sum_{t=1}^{T} r^i\big(a_t^i, a_t^{-i}, z_t\big). \tag{1}$$

The contextual regret compares the cumulative reward obtained throughout the game with the one achievable by the *best fixed policy* in hindsight, i.e., had player $i$ known the sequence $\{z_t, a_t^{-i}\}_{t=1}^{T}$ of contexts and opponents' actions ahead of time, as well as the reward function $r^i(\cdot)$. Crucially, $R_c(T)$ sets a stronger benchmark than competing only with the best fixed action $a \in \mathcal{A}^i$ and captures the fact that players should use the revealed context information to improve their performance. A strategy is *no-regret* for player $i$ if $R_c^i(T)/T \to 0$ as $T \to \infty$.

Contextual games generalize the class of standard (non-contextual) repeated games, allowing the game to change from round to round due to a potentially different context $z_t$ (we recover the standard repeated games setup and regret definition by assuming $z_t = z_0$ for all $t$). In Section 4 we define new notions of *equilibria* and *efficiency* for such games and show that the contextual regret defined in (1), besides measuring individual players' performance, has a close connection with game equilibria and efficiency. First, however, motivated by these considerations, we focus on the individual perspective of a generic player $i$ and seek to derive suitable no-regret strategies. In this regard, the algorithms and guarantees presented in the next section do not rely on the other players complying with any pre-specified rule, but consider the worst case over the opponents' actions $a_t^{-i}$ (also, potentially chosen as a function of the observed game data). To simplify notation, we denote player $i$'s reward function with $r(\cdot)$, unless otherwise specified.

**Regularity assumptions.** Even with a single known context, achieving no-regret is impossible unless we make further assumptions on the game [11]. We assume the action set $\mathcal{A}^i$ is finite with $|\mathcal{A}^i| = K$. We consider a generic set $\mathcal{Z} \subseteq \mathbb{R}^c$ and make no assumptions on how contexts are generated (they could be adversarially chosen by Nature, possibly as a function of past game rounds). In Section 3.3, however, we consider a special case where contexts are sampled i.i.d. from a static distribution $\zeta$. Our next regularity assumptions concern the reward function $r(\cdot)$.

Let $\mathcal{D} := \mathcal{A} \times \mathcal{Z}$. We assume the unknown function $r(\cdot)$ has a bounded norm in a Reproducing Kernel Hilbert Space (RKHS) associated with a positive-semidefinite kernel function $k : \mathcal{D} \times \mathcal{D} \to [-1, 1]$. Kernel $k(\cdot, \cdot)$ measures the similarity between two different context-action pairs, and the norm $\|r\|_k = \sqrt{\langle r, r\rangle_k}$ measures the smoothness of $r$ with respect to $k$. This is a standard non-parametric assumption used, e.g., in Bayesian optimization [37] and recently exploited to model correlations in repeated games [32]. It encodes the fact that similar action profiles (e.g., similar network's occupancy profiles in a routing game) lead to similar rewards (travel times), and allows player $i$ to generalize experience to non-played actions and, in our case, unseen contexts. Popularly used kernels include polynomial, Squared Exponential (SE), and Matérn kernels [29], while composite kernels [22] can also be used to encode different dependences of $r$ on $a^i$, $a^{-i}$, and context $z$.

**Feedback model.** We assume player $i$ receives a *noisy* bandit observation $\tilde{r}_t = r(a_t^i, a_t^{-i}, z_t) + \epsilon_t$ of the reward at each round, where $\epsilon_t$ is $\sigma$-sub-Gaussian (i.e., $\mathbb{E}[\exp(\alpha\epsilon_t)] \le \exp(\alpha^2\sigma^2/2), \forall\alpha \in \mathbb{R}$) and independent over time. Moreover, we assume, similar to [32], that at the *end* of each round, player $i$ also observes the actions $a_t^{-i}$ chosen by the other players. The latter assumption will allow player $i$ to achieve improved performance compared to the standard bandit feedback. In some applications (e.g., aggregative games such as traffic routing), it is only sufficient to observe an aggregate function of $a_t^{-i}$.

## 3 Algorithms and Guarantees

From the perspective of player $i$, playing a contextual game corresponds to an *adversarial contextual bandit problem* (see, e.g., [8, Chapter 4]) where, at each round, context $z_t$ is revealed, an adversary picks a reward function $r_t(\cdot, z_t) : \mathcal{A}^i \to [0, 1]$ and player $i$ obtains reward $r_t(a_t^i, z_t)$. Therefore, player $i$ could in principle use existing adversarial contextual bandits algorithms to achieve no-regret. Such algorithms come with different regret guarantees depending on the assumptions made but, importantly, incur a regret which scales poorly with the size of the action space $\mathcal{A}^i$. This is because they use high-variance estimators to estimate the rewards of non-played actions, i.e., the so-called *full information* feedback. Also, some of these algorithms assume parametric (e.g., linear [28]) dependence of the rewards $r_t(\cdot, z_t)$ on the context $z_t$ and hence cannot deal with our general game structure.

---

**Algorithm 1** The C.GP-MW (meta) algorithm

---

**Require:** Finite set $\mathcal{A}^i$ of $K$ actions, kernel $k$, learning rates $\{\eta_t\}_{t \geq 0}$, confidence levels $\{\beta_t\}_{t \geq 0}$.
1: **for** $t = 1, \ldots, T$ **do**
    /* Nature chooses context $z_t$                                                      /*
2:       Observe context $z_t$
3:       Compute distribution $p_t(z_t) \in \Delta^K$ using: $z_t$, $\eta_t$, and $\{\mathrm{ucb}_\tau(\cdot), a_\tau^{-i}, z_\tau\}_{\tau=1}^{t-1}$.
4:       Sample action $a_t^i \sim p_t(z_t)$
    /* Simultaneously, opponents choose their actions $a_t^{-i}$                    /*
5:       Observe noisy reward $\tilde{r}_t$ and opponents' actions $a_t^{-i}$       // $\tilde{r}_t = r(a_t^i, a_t^{-i}, z_t) + \epsilon_t$
6:       Use the observed data to update $\mathrm{ucb}_t(\cdot)$ according to (2) and (3).

---

Instead, we exploit the fact that in a contextual game the rewards obtained at different times are *correlated* through the reward function $r(\cdot)$ (i.e., contextual games correspond to the specific contextual bandit problem where $r_t(\cdot, z_t) = r(\cdot, a_t^{-i}, z_t)$ for all $t$). This fact, together with our feedback model, allows player $i$ to use past game data to obtain (with increasing confidence) an estimate of the reward function $r$ and use it to emulate the full-information feedback.

**RKHS regression.** Using past game data $\{a_\tau^i, a_\tau^{-i}, z_\tau, \tilde{r}_\tau\}_{\tau=1}^t$, standard kernel ridge regression [29] can be used to compute posterior mean and corresponding variance estimates of the reward function $r(\cdot)$. For any $x = (a, a^{-i}, z) \in \mathcal{D}$, and regularization parameter $\lambda > 0$, they can be obtained as:

$$\mu_t(x) = \mathbf{k}_t(x)^T \left(\mathbf{K}_t + \lambda \mathbf{I}_t\right)^{-1} \mathbf{y}_t, \quad \sigma_t^2(x) = k(x,x) - \mathbf{k}_t(x)^T \left(\mathbf{K}_t + \lambda \mathbf{I}_t\right)^{-1} \mathbf{k}_t(x), \quad (2)$$

where $\mathbf{k}_t(x) = \left[k\left(x_j, x\right)\right]_{j=1}^t$, $\mathbf{y}_t = \left[\tilde{r}_j\right]_{j=1}^t$, and $\mathbf{K}_t = \left[k\left(x_j, x_{j'}\right)\right]_{j,j'}$ is the kernel matrix. Moreover, such estimates can be used to build the upper confidence bound function:

$$\mathrm{ucb}_t(\cdot) := \min\{\mu_t(\cdot) + \beta_t \sigma_t(\cdot), 1\}, \quad (3)$$

where $\beta_t$ is a confidence parameter, and the function is truncated at 1 since $r(x) \in [0,1]$ for all $x \in \mathcal{D}$. A standard result from [37] shows that $\beta_t$ can be chosen such that $r(x) \in [\mu_t(x) + \beta_t \sigma_t(x), \mu_t(x) - \beta_t \sigma_t(x)]$ with high probability for any $x \in \mathcal{D}$ (see Lemma 7 in the Appendix). The function $\mathrm{ucb}_t(\cdot)$ hence represents an optimistic estimate of $r(\cdot)$ and can be used by player $i$ to emulate the full-information feedback. We outline our (meta) algorithm C.GP-MW in Algorithm 1.

C.GP-MW extends and generalizes the recently proposed GP-MW [32] algorithm to play repeated games, to the case where contextual information is available to the players and the goal is to compete with the optimal policy in hindsight. At each time step, after observing context $z_t$ the algorithm computes a distribution $p_t(z_t) \in \Delta^K$, where $\Delta^K$ is the $K$-dimensional simplex, and samples an action from it. At the same time, the algorithm uses the observed game data to construct upper confidence bound functions of the player's rewards using (3). Such functions, together with the observed context $z_t$ are used to compute the distribution $p_t(z_t)$ at each round. Note that C.GP-MW is a well-defined algorithm after we specify the rule used to compute $p_t(z_t)$ (line 3 of Algorithm 1). We left such rule unspecified, as we will specialize it to different settings throughout this section.

The regret bounds obtained in this section depend on the so-called *maximum information gain* [37] about the unknown function $r(\cdot)$ from $T$ noisy observations, defined as:

$$\gamma_T := \max_{\{x_t\}_{t=1}^T} 0.5 \log \det(\mathbf{I}_T + \mathbf{K}_T / \lambda).$$

This quantity represents a sample-complexity parameter which, importantly, for popularly used kernels does not grow with the number of actions $K$ but only with the dimension $d$ of the domain $\mathcal{D}$. It can be bounded analytically as, e.g., $\gamma_T \leq \mathcal{O}(d \log T)$ and $\gamma_T \leq \mathcal{O}(\log(T)^{d+1})$ for squared exponential and linear kernels, respectively [37]. Moreover, we remark that although in the worst case $d$ grows linearly with the number of players $N$, in many applications (such as the traffic routing game considered in Section 5) the reward function $r(\cdot)$ depends only on some aggregate function of the opponents' actions $a^{-i}$ and therefore $d$ is independent from the number of players in the game.

## 3.1 Finite (small) number of contexts

When the context set $\mathcal{Z}$ is finite, a basic version of C.GP-MW achieves a high-probability regret bound of $\mathcal{O}(\sqrt{T|\mathcal{Z}| \log K} + \gamma_T \sqrt{T})$: We simply maintain a distribution $p_t(z) \in \Delta^K$ for each context $z$ and update it only when $z$ is observed, using the Multiplicative Weights (MW) method [26]. We formally introduce and study such strategy in Appendix A.1. In the same setting, and with standard

---
**Strategy 2** Exploit similarity across contexts
---
1: **Set** Radius $\epsilon > 0$, $\mathcal{C} = \{z_1\}$, and let $p_1(z_1)$ be the uniform distribution.
2: **for** $t = 2, \ldots, T$ **do**
3:     Observe context $z_t$ and let $z'_t = \arg\min_{z \in \mathcal{C}} \|z_t - z\|_1$
4:     **if** $\|z_t - z'_t\|_1 > \epsilon$ **then**
5:         Add $z_t$ to the set $\mathcal{C}$, set $z'_t = z_t$, and let $p_t(z_t)$ be the uniform distribution
6:     **else**
$$p_t(z_t)[a] \propto \exp\left( \eta_t \cdot \sum_{\tau=1}^{t-1} \mathrm{ucb}_\tau(a, a_\tau^{-i}, z_\tau) \cdot \mathbb{1}\{z'_\tau = z'_t\} \right) \qquad a = 1, \ldots, K. \tag{4}$$
---

bandit feedback, the $\mathcal{S}$-EXP3[8] algorithm achieves regret $\mathcal{O}(\sqrt{T|\mathcal{Z}|K\log K})$, which has a worse dependence on the number of actions $K$. These regret bounds, however, are appealing only when the set $\mathcal{Z}$ has low cardinality, and become worthless otherwise. Intuitively, this is because no information is shared across different contexts and each context is treated independently from each other.

## 3.2  Exploit similarity across contexts

For large or even infinite $\mathcal{Z}$, we want to exploit the fact that similar contexts should lead to similar performance and take this into account when computing the action distribution $p_t(z_t)$. We capture this fact by assuming the optimal policy in hindsight $\pi^\star = \arg\max_{\pi \in \Pi^i} \sum_{t=1}^T r(\pi(z_t), a_t^{-i}, z_t)$ is $L_p$-Lipschitz:
$$|\pi^\star(z_1) - \pi^\star(z_2)| \leq L_p \|z_1 - z_2\|_1, \quad \forall z_1, z_2 \in \mathcal{Z}.$$
 Moreover, we assume $\mathcal{Z} \subseteq [0,1]^c$ to obtain a scale-free regret bound, and that the reward function $r(\cdot)$ is $L_r$-Lipschitz with respect to the decision set $\mathcal{A}^i$, i.e., $|r(a_1, a^{-i}, z) - r(a_2, a^{-i}, z)| \leq L_r \|a_1 - a_2\|_1, \forall a_1, a_2 \in \mathcal{A}^i, \forall (a^{-i}, z)$, which is readily satisfied for most kernels [12, Lemma 1].

These assumptions allow player $i$ to share information across different, but similar, contexts to improve the performance. This can be done by using the online Strategy 2 to compute $p_t(z_t)$ at each round (Line 3 in Algorithm 1).  Such strategy consists of building, in a greedy fashion as new contexts are revealed, an $\epsilon$-net [23] of the context space $\mathcal{Z}$, similarly to the algorithm by [20] for online convex optimization: At each time $t$, either a new L1-ball centered at $z_t$ is created or $z_t$ is assigned to the closest ball. In the latter case, $p_t(z_t)$ is computed via a MW rule using the sequence of $\mathrm{ucb}_\tau(\cdot)$ functions for those time steps $\tau < t$ that $z_\tau$ belongs to such ball. Note that Strategy 2 can also be implemented recursively, by maintaining a probability distribution for each new ball and updating only the one corresponding to the ball $z_t$ belongs to. The radius $\epsilon$ is a tunable parameter, which can be set as follows.

**Theorem 1.** *Fix $\delta \in (0,1)$ and assume $\|r^i\|_k \leq B$, $\pi^\star$ is $L_p$-Lipschitz, and $r^i$ is $L_r$-Lipschitz in $\mathcal{A}^i$. If player $i$ plays according to* C.GP-MW *using Strategy 2 with $\lambda \geq 1$, $\beta_t = B + \sigma\lambda^{-1/2}\sqrt{2(\gamma_{t-1} + \log(2/\delta))}$, $\eta_t = 2\sqrt{\log K / \sum_{\tau=1}^t \mathbb{1}\{z'_\tau = z'_t\}}$, and $\epsilon = (L_r L_p)^{-\frac{2}{c+2}} T^{-\frac{1}{c+2}}$, then with probability at least $1 - \delta$,*
$$R_c^i(T) \leq 2(L_r L_p)^{\frac{c}{c+2}} T^{\frac{c+1}{c+2}} \sqrt{\log K} + \sqrt{0.5T\log(2/\delta)} + 4\beta_T \sqrt{\gamma_T \lambda T}.$$

Compared to Section 3.1, the obtained regret bound is now independent of the size of $\mathcal{Z}$, although its sublinear dependence on $T$ degrades with the contexts' dimension $c$. The additive $\mathcal{O}(\beta_T\sqrt{\gamma_T T})$ term represents the cost of learning the reward function $r(\cdot)$ online. Note that even if $r(\cdot)$ was *known*, and hence full-information feedback was available, the $\mathcal{O}(\sqrt{T|\mathcal{Z}|})$ and $\mathcal{O}(T^{\frac{c+1}{c+2}})$ rates obtained so far are shown optimal in their respective settings, i.e., when $\mathcal{Z}$ is finite [8] or when the discussed Lipschitz assumptions are satisfied [20]. An interesting future direction is to understand whether more refined bounds can be derived as a function of the contexts' sequence using adaptive partitions as proposed by [36]. In the next section we show that significantly improved guarantees are achievable when contexts are i.i.d. samples from a static distribution.

Finally, we remark that all the discussed computations are efficient, as they do not iterate over the set of policies $\Pi^i$ (which has exponential size). Improved regret bounds can be obtained if this requirement is relaxed, e.g., assuming a finite pool of policies [3], or a value optimization oracle [39]. We believe such results are complementary to our work and can be coupled with our RKHS game assumptions.

### 3.3 Stochastic contexts and non-reactive opponents

In this section, we consider a special case where contexts are i.i.d. samples from a static distribution $\zeta$, i.e., $z_t \sim \zeta$ for $t = 1, \ldots, T$. Importantly, we consider the realistic case in which player $i$ does neither know, nor can sample from, such distribution. Moreover, we focus on the setting where the opponents' decisions $a_t^{-i}$ are *not* based on the current realization of $z_t$, but can only depend on the history of the game. Examples of such a setting are games where the context $z_t$ represents 'private' information for player $i$ (e.g., in Bayesian games, $z_t$ can represent player $i$'s *type* [19]), or where $z_t$ is only relevant to player $i$ and hence the opponents have no reason to decide based on it.

In this case, we analyze the following strategy to compute $p_t(z_t)$ at each round (Line 3 in Algorithm 1):

$$p_t(z_t)[a] \propto \exp\left(\eta_t \cdot \sum_{\tau=1}^{t-1} \mathrm{ucb}_\tau(a, a_\tau^{-i}, z_t)\right) \qquad a = 1\ldots, K. \tag{5}$$

Crucially, the distribution $p_t(z_t)$ is now computed using the *whole sequence* of past $\mathrm{ucb}_\tau$ functions, evaluated at context $z_t$, regardless of whether $z_t$ was observed in the past. Hence, while according to rule (4) – and most of the MW algorithms – $p_t(z_t)$ can be updated in a recursive manner, using rule (5) such distribution is re-computed at each round after observing $z_t$ (this requires storing the previous $\mathrm{ucb}_\tau$ functions, or re-computing them using (2) at each round).[1] Such strategy exploits the stochastic assumption on the contexts and reduces the contextual game to a set of auxiliary games, one for each context. This idea was recently used also by [5, 28] in the finite and linear contextual bandit setting, respectively, while we specialize it to repeated games coupled with our RKHS assumptions.

The next theorem provides a pseudo-regret bound for C.GP-MW when using strategy (5), i.e., we bound the quantity $\mathbb{E}R_c^i(T, \pi)$ (expectation with respect to the contexts' sequence and the randomization of C.GP-MW), where $R_c^i(T, \pi)$ is the regret with respect to a generic policy $\pi \in \Pi$. Note that the pseudo-regret is smaller than the expected contextual regret $\mathbb{E}R_c^i(T)$ which, however, is proven to grow linearly with $T$ when $\mathcal{Z}$ is sufficiently large [5]. Nevertheless, [5, Theorem 22] shows that $|\mathbb{E}R_c^i(T, \pi) - \mathbb{E}R_c^i(T)|$ can be bounded assuming each context occurs sufficiently often.

**Theorem 2.** *Fix $\delta \in (0,1)$ and assume $\|r^i\|_k \leq B$ and $z_t \sim \zeta$ for all $t$. Moreover, assume the opponents cannot observe the current context $z_t$. If player $i$ plays according to C.GP-MW using strategy (5) with $\lambda \geq 1$, $\beta_t = B + \sigma\lambda^{-1/2}\sqrt{2(\gamma_{t-1} + \log(1/\delta))}$ and $\eta_t = \sqrt{(8\log K)/T}$, then with probability at least $1 - \delta$,*

$$\sup_{\pi \in \Pi^i} \mathbb{E}\left[\sum_{t=1}^T r\big(\pi(z_t), a_t^{-i}, z_t\big) - \sum_{t=1}^T r\big(a_t, a_t^{-i}, z_t\big)\right] \leq \sqrt{0.5 T \log K} + 4\beta_T\sqrt{\gamma_T \lambda T},$$

*where expectation is with respect to both the contexts' sequence and the randomization of C.GP-MW.*

The above guarantee significantly improves upon the ones obtained in the previous sections, as it does not depend on the context space $\mathcal{Z}$, and matches the regret of GP-MW in non-contextual games. It should be compared with the $\mathcal{O}(\sqrt{TK\log K})$ guarantee of [5] which assumes rewards for the non-revealed contexts are also observed, and the bandit $\mathcal{O}(\sqrt{cTK\log K})$ pseudo-regret of [28] assuming a linear dependence between contexts and rewards and a *known* contexts distribution. Exploiting our game assumptions, C.GP-MW's performance decreases only logarithmically with $K$, relies on a more realistic feedback model than [5] and can deal with more complex rewards structures than [28].

## 4 Game Equilibria and Efficiency

In this section, we introduce new notions of equilibria and efficiency for contextual games. We recover game-theoretic learning results [18, 31] showing that equilibria end efficiency (as defined below) can be approached when players minimize their contextual regret.

### 4.1 Contextual Coarse Correlated Equilibria

A typical solution concept of multi-player *static* games is the notion of Coarse Correlated Equilibria (CCEs) (see, e.g., [31, Section 3.1]). CCEs include Nash equilibria and have received increased attention because of their amenability to learning: a fundamental result from [18] shows that CCEs can be approached by decentralized no-regret dynamics, i.e., when each player uses a no-regret algorithm. These results, however, are not applicable to contextual games, where suitable notions of

equilibria should capture the fact that players can observe the current context before playing. To cope with this, we define a notion of CCEs for contextual games, denoted as *contextual CCE (c-CCE)*.

**Definition 3** (Contextual CCE). *Consider a contextual game described by contexts $z_1, \ldots, z_T$. Let $\Pi^i$ be the set of all policies $\pi : \mathcal{Z} \to \mathcal{A}^i$ for player $i$, and $\mathcal{A}$ be the joint space of actions $\mathbf{a} = (a^i, a^{-i})$. A contextual coarse-correlated equilibrium (c-CCE) is a policy $\rho : \mathcal{Z} \to \Delta^{|\mathcal{A}|}$ such that:*

$$\frac{1}{T} \sum_{t=1}^{T} \underset{\mathbf{a} \sim \rho(z_t)}{\mathbb{E}} r^i(\mathbf{a}, z_t) \geq \frac{1}{T} \sum_{t=1}^{T} \underset{\mathbf{a} \sim \rho(z_t)}{\mathbb{E}} r^i(\pi(z_t), a^{-i}, z_t) \quad \forall \pi \in \Pi^i, \quad \forall i = 1, \ldots, N. \quad (6)$$

As opposed to CCEs (which are elements of $\Delta^{|\mathcal{A}|}$), a c-CCE is a *policy* $\rho : \mathcal{Z} \to \Delta^{|\mathcal{A}|}$ from which no player has incentive to deviate looking at the time-averaged expected reward. In other words, suppose there is a trusted device that, for any context $z_t$, samples a joint action from $\rho(z_t)$, where $\rho$ is a c-CCE. Then, in expectation, each player is better off complying with such device, instead of using any other $\pi : \mathcal{Z} \to \mathcal{A}^i$. We say that $\rho$ is a $\epsilon$-c-CCE if inequality (6) is satisfied up to a $\epsilon \in \mathbb{R}_+$ accuracy. Finally, we remark that c-CCEs reduce to CCEs in case $z_t = z_0$ for all $t$.

**Example (c-CCEs in traffic routing)** In traffic routing applications, the trusted device can be a routing system (e.g., a maps server) which, given current weather, traffic conditions, or other contextual information, decides on a route for each user. If such routes are sampled according to a c-CCE, then each user is better-off complying with such device to ensure a minimum expected travel time.

The next proposition shows that, similarly to CCEs in static games, c-CCEs can be approached whenever players minimize their contextual regrets. Hence, it provides a fully decentralized and efficient scheme for computing $\epsilon$-c-CCEs. To do so, we define the notion of *empirical policy* at round $T$ as follows. After $T$ game rounds, let $\mathcal{Z}_T$ be the set of all the distinct observed contexts. Then, the empirical policy at round $T$ is the policy $\rho_T : \mathcal{Z} \to \Delta^{|\mathcal{A}|}$ such that, for each $z \in \mathcal{Z}_T$, $\rho_T(z)$ is the empirical distribution of played actions when context $z$ was revealed, while for the unseen contexts $z \in \mathcal{Z} \setminus \mathcal{Z}_T$, $\rho_T(z)$ is an arbitrary (e.g., uniform) distribution.

**Proposition 4** (Finite-time approximation of c-CCEs). *After $T$ game rounds, let $R_c^i(T)$'s denote the players' contextual regrets and $\rho_T$ be the empirical policy at round $T$. Then, $\rho_T$ is a $\epsilon$-c-CCE of the played contextual game with $\epsilon \leq \max_{i \in \{1, \ldots, N\}} R_c^i(T)/T$.*

Proposition 4 implies that, as $T \to \infty$, if players use vanishing contextual regret algorithms (such as the ones discussed in Section 3), then the empirical policy $\rho_T$ converges to a c-CCE of the contextual game. When contexts are stochastic, i.e., $z_t \sim \zeta$ for all $t$, an alternative notion of c-CCE can be defined by considering the *expected* context realization. We treat this case in Appendix B.2 and prove similar finite-time and asymptotic convergence results using standard concentration arguments.

## 4.2 Approximate Efficiency

The efficiency of an outcome $\mathbf{a} \in \mathcal{A}$ in a non-contextual game, i.e., for a fixed context $z_0$, can be quantified as the distance between the *social welfare* $\Gamma(\mathbf{a}, z_0) := \sum_{i=1}^{N} r^i(\mathbf{a}, z_0)$ (where $:=$ is sometimes replaced by $\geq$ if the reward of the game authority is also considered), and the optimal welfare $\max_{\mathbf{a}} \Gamma(\mathbf{a}, z_0)$. Optimal welfare is typically not achieved as players are self-interested agents aiming at maximizing their individual rewards, instead of $\Gamma$. Nevertheless, main results of [6, 31] show that such efficiency loss can be bounded whenever players minimize their regrets and the game is $(\lambda, \mu)$-*smooth*, i.e., if for any pair of outcomes $\mathbf{a}_1 = (a_1^1, \ldots, a_1^N)$ and $\mathbf{a}_2 = (a_2^1, \ldots, a_2^N)$, it satisfies:

$$\sum_{i=1}^{N} r^i(a_2^i, a_1^{-i}, z_0) \geq \lambda \cdot \Gamma(\mathbf{a}_2, z_0) - \mu \cdot \Gamma(\mathbf{a}_1, z_0). \quad (7)$$

Examples of smooth games are routing games with polynomial delay functions, several classes of auctions, submodular welfare games, and many more (see, e.g., [40, 31, 33] and references therein).

In contextual games, on the other hand, a different context $z_t$ describes the game at each round, and hence the social welfare $\Gamma(\mathbf{a}, z_t)$ of an outcome $\mathbf{a} \in \mathcal{A}$ depends on the specific context realization. The efficiency of a contextual game can therefore be quantified by the optimal *contextual* welfare:

**Definition 5** (Optimal contextual welfare).

$$\text{OPT} = \max_{\pi^1 \in \Pi^1, \ldots, \pi^N \in \Pi^N} \frac{1}{T} \sum_{t=1}^{T} \Gamma(\pi^1(z_t), \ldots, \pi^N(z_t), z_t). \quad (8)$$

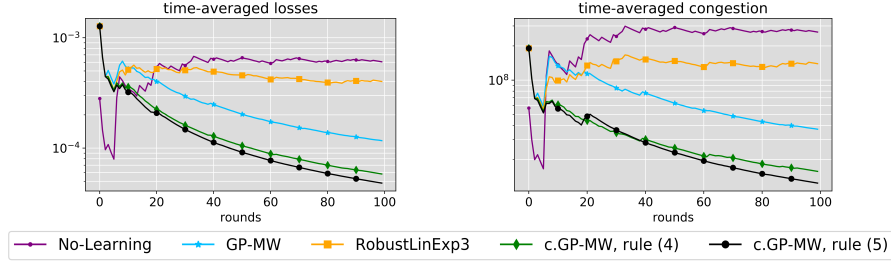

Figure 1: Time-averaged losses (Left) and network congestion (Right), when agents use different routing strategies (average over 5 runs). C.GP-MW leads to reduced losses and congestion compared to the other baselines.

Equation (8) generalizes the optimal welfare for non-contextual games, and sets the stronger benchmark of finding the *policies* (instead of static actions) mapping contexts to actions which maximize the time-averaged social welfare. In routing games, for instance, it corresponds to finding the best routes for the agents as a function of current traffic conditions. As shown in our experiments (Section 5), such policies can significantly reduce the network's congestion compared to finding the best static routes. The next proposition generalizes the well-known results of [31], showing that in a smooth contextual game such optimal welfare can be approached when players minimize their contextual regret. First, we note that a contextual game can satisfy the smoothness condition (7) for different constants $\lambda, \mu$, depending on the context $z_t$, and hence we use the notation $\lambda(z_t), \mu(z_t)$ to highlight their dependence.

**Proposition 6** (Convergence to approximate efficiency). *Let $R_c^i(T)$'s be the players' contextual regrets, and assume the game is $(\lambda(z_t), \mu(z_t))$-smooth at each time t. Then,*

$$\frac{1}{T} \sum_{t=1}^{T} \Gamma(a_t^1, \dots, a_t^N, z_t) \geq \frac{\bar{\lambda}}{1 + \bar{\mu}} \text{ OPT } - \frac{1}{1 + \bar{\mu}} \sum_{i=1}^{N} \frac{R_c^i(T)}{T},$$

*where $\bar{\lambda} = \max_{t \in \{1, \dots, T\}} \lambda(z_t)$ and $\bar{\mu} = \min_{t \in \{1, \dots, T\}} \mu(z_t)$.*

The approximation factor $\bar{\lambda}/(1 + \bar{\mu})$ (also known as Price of Total Anarchy [6]) depends on the constants $\bar{\lambda}$ and $\bar{\mu}$, which in our case represent the 'worst-case' smoothness of the game. We remark however that game smoothness is not necessarily context-dependent, e.g., routing games (such as the one considered in the next section) are smooth regardless of the network's size and capacities [31].

## 5  Experiments - Contextual Traffic Routing Game

We consider a *contextual* routing game on the traffic network of Sioux-Falls, a directed graph with 24 nodes and 76 edges, with the same game setup of [32] (network data and congestion model are taken from [25], while Appendix C gives a complete description of our experimental setup). There are $N = 528$ agents in the network. Each agent wants to send $d_i$ units from a given origin to a given destination node in minimum time, and can choose among $K = 5$ routes at each round. The traveltime of an agent depends on the routes chosen by the other agents (if all choose the same route the network becomes highly congested) as well as the network's capacity at round $t$. We let $x_t^i \in \mathbb{R}^{76}$ represent the route chosen by agent $i$ at round $t$, where $x_t^i[e] = d_i$ if edge $e$ belongs to such route, and $x_t^i[e] = 0$ otherwise. Moreover, we let context $z_t \in \mathbb{R}_+^{76}$ represent the capacity of the network's edges at round $t$ (capacities are i.i.d. samples from a fixed distribution, see Appendix C). Then, agents' rewards can be written as $r^i(x_t^i, x_t^{-i}, z_t) = -\sum_{e=1}^{76} x_t^i[e] \cdot t_e(x_t^i + x_t^{-i}, z_t[e])$, where $x_t^{-i} = \sum_{j \neq i} x_t^j$ and $t_e(\cdot)$'s are the edges' traveltime functions, which are unknown to the agents. Note that, strictly speaking, agent $i$'s reward depends only on the entries $x_t^i[e], x_t^{-i}[e], z_t[e]$ for $e \in E^i$, where $E^i$ is the subset of edges that agent $i$ could potentially traverse. According to our model, we assume agent $i$ observes context $\{z_t[e], e \in E^i\}$ and, at end of each round, the edges' occupancies $\{x_t^{-i}[e], e \in E^i\}$.

We let each agent select routes according to C.GP-MW (using rule (4) or (5)) and compare its performance with the following baselines: 1) No-Learning, i.e., agents select the shortest free-flow routes at each round, 2) GP-MW [32] which neglects the observed contexts, and 3) ROBUSTLINEXP3 [28] for contextual linear bandits, which is robust to model misspecification but does not exploit the correlation in the game (also, it requires knowing the contexts' distribution). To run C.GP-MW we use the composite kernel $k(x_t^i, x_t^{-i}, z_t) = k_1(x_t^i) * k_2((x_t^i + x_t^{-i})/z_t)$, while for GP-MW the kernel $k(x_t^i, x_t^{-i}, z_t) = k_1(x_t^i) * k_2(x_t^i + x_t^{-i})$, where $k_1, k_2$ are linear and polynomial kernels respectively.

We set $\eta_t$ according to Theorems 1 and 2, and $\beta_t = 2.0$ (theoretical values for $\beta_t$ are found to be overly conservative [37, 32]). For rule (4) we set $\epsilon = 30|E^i|$. Figure 1 shows the time-averaged losses (i.e., traveltimes scaled in $[0, 1]$ and averaged over all the agents), which are inversely proportional to the game welfare, and the resulting network's congestion (computed as in Appendix C). We observe, in line to what discussed in Section 4, that minimizing individual regrets the agents increase the game welfare (this is expected as routing games are smooth [30]). Moreover, when using C.GP-MW agents exploit the observed contexts and correlations, and achieve significantly more efficient outcomes and lower congestion levels compared to the other baselines. We also observe strategy (5) outperforms strategy (4) in our experiments. This can be explained by the contexts being stochastic and, also, since each agent $i$ is only influenced by the coordinates $z_t[e]$ of the relevant edges $e \in E^i$.

## 6 Conclusions

We have introduced the class of contextual games, a type of repeated games described by contextual information at each round. Using kernel-based regularity assumptions, we modeled the correlation between different contexts and game outcomes, and proposed novel online algorithms that exploit such correlations to minimize the players' contextual regret. We defined the new notions of contextual Coarse Correlated Equilibria and optimal contextual welfare and showed that these can be approached when players have vanishing contextual regret. The obtained results were validated in a traffic routing experiment, where our algorithms led to reduced travel times and more efficient outcomes compared to other baselines that do not exploit the observed contexts or the correlation present in the game.

## Broader Impact

As systems using machine learning get deployed more and more widely, these systems increasingly interact with each other. Examples range from road traffic over auctions and financial markets, to robotic systems. Understanding these interactions and their effects for individual participants and the reliability of the overall system becomes ever more important. We believe our work contributes positively to this challenge by studying principled algorithms that are efficient, while converging to suitable, and often efficient, equilibria.

### Acknowledgments

This work was gratefully supported by the Swiss National Science Foundation, under the grant SNSF 200021_172781, by the European Union's ERC grant 815943, and ETH Zürich Postdoctoral Fellowship 19-2 FEL-47.

## Footnotes

[1] We note that strategy (5) can be implemented recursively in case the contexts' set $\mathcal{Z}$ is known and finite.

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
