[Supplementary Material]

# Supplementary Material

## Contextual Games: Multi-Agent Learning with Side Information

**Pier Giuseppe Sessa, Ilija Bogunovic, Andreas Krause, Maryam Kamgarpour (NeurIPS 2020)**

## A  Supplementary Material for Section 3

The theoretical guarantees obtained in Section 3 rely on the following two main lemmas. The first lemma is from [2] and shows that given the previously observed rewards, contexts, and players' actions, the reward function of player $i$ belongs (with high probability) to the interval $[\mu_t(\cdot, \cdot) \pm \beta_t \sigma_t(\cdot, \cdot)]$, for a carefully chosen confidence parameter $\beta_t \geq 0$.

**Lemma 7.** *Let $r \in \mathcal{H}_k$ such that $\|r\|_k \leq B$ and consider the kernel-ridge regression mean and standard deviation estimates $\mu_t(\cdot)$ and $\sigma_t(\cdot)$, with regularization constant $\lambda > 0$. Then for any $\delta \in (0,1)$, with probability at least $1 - \delta$, the following holds simultaneously over all $x \in \mathcal{D}$ and $t \geq 1$:*

$$|\mu_t(x) - r(x)| \leq \beta_t \sigma_t(x),$$

*where $\beta_t = B\lambda^{-1/2} + \sigma\lambda^{-1}\sqrt{2\log\left(\frac{1}{\delta}\right) + \log(\det(I_t + K_t/\lambda))}$.*

Therefore, according to Lemma 7, the function $\mathrm{ucb}_t$ defined in (3) represents a valid upper confidence bound for the rewards obtained by player $i$.

The second main lemma concerns the properties of the Multiplicative Weights (MW) update method [26], which is used as a subroutine in our algorithms to compute the action distribution $p_t(z_t)$ (Line 3 of Algorithm 1) at each round. Its proof follows from standard online learning arguments equivalently to, e.g., [27, Proposition 1].

**Lemma 8.** *Consider a sequence of functions $g_1(\cdot), \ldots, g_T(\cdot) \in [0,1]^K$ and let $p_t$'s be the distributions computed using the MW rule:*

$$p_t[a] \propto \exp\left(\eta_t \cdot \sum_{\tau=1}^{t-1} g_\tau(a)\right) \qquad a = 1, \ldots, K, \tag{9}$$

*where $p_1$ is initialized as the uniform distribution. Then, provided that $\{\eta_t\}_{t=1}^T$ is a decreasing sequence, for any action $a^\star \in \{1, \ldots, K\}$:*

$$\sum_{t=1}^T g_t(a^\star) - \sum_{t=1}^T \sum_{a=1}^K p_t[a] \cdot g_t(a) \leq \frac{\log K}{\eta_T} + \frac{\sum_{t=1}^T \eta_t}{8}.$$

### A.1  The case of a finite (and small) number of contexts

In this section we consider the simple case of a finite (and small cardinality) set of contexts $\mathcal{Z}$. In such a case, a high-probability regret bound of $\mathcal{O}(\sqrt{T|\mathcal{Z}|\log K} + \gamma_T\sqrt{T})$ can be achieved when C.GP-MW is run with the following strategy:

$$p_t(z_t)[a] \propto \exp\left(\eta_t \cdot \sum_{\tau=1}^{t-1} \mathrm{ucb}_\tau(a, a_\tau^{-i}, z_\tau) \cdot \mathbb{1}\{z_\tau = z_t\}\right) \qquad a = 1, \ldots, K. \tag{10}$$

That is, $p(z_t)$ is computed using the sequence of previously computed upper confidence bound functions for the game rounds in which the specific context $z_t$ was revealed.

**Theorem 9.** *Fix $\delta \in (0,1)$ and assume $\|r^i\|_{k^i} \leq B$. If player $i$ plays according to C.GP-MW using strategy (10) with $\lambda \geq 1$, $\beta_t = B + \sigma\lambda^{-1/2}\sqrt{2(\gamma_{t-1} + \log(2/\delta))}$, and $\eta_t = 2\sqrt{\log K / \sum_{\tau=1}^t \mathbb{1}\{z_\tau = z_t\}}$, then with probability at least $1 - \delta$,*

$$R_c^i(T) \leq \sqrt{T|\mathcal{Z}|\log K} + \sqrt{0.5T\log(2/\delta)} + 4\beta_T\sqrt{\gamma_T\lambda T}.$$

*Proof.* Let $\pi^\star = \arg\max_{\pi \in \Pi^i} \sum_{t=1}^{T} r\big(\pi(z_t), a_t^{-i}, z_t\big)$. Our goal is to bound $R_c^i(T) = \sum_{t=1}^{T} r\big(\pi^\star(z_t), a_t^{-i}, z_t\big) - r\big(a_t^i, a_t^{-i}, z_t\big)$ with high probability.

By conditioning on the event of the confidence lemma (Lemma 7) holding true, we can state that, with probability at least $1 - \delta/2$,

$$
\begin{aligned}
R_c^i(T) &= \sum_{t=1}^{T} r\big(\pi^\star(z_t), a_t^{-i}, z_t\big) - r\big(a_t^i, a_t^{-i}, z_t\big) \\
&\leq \sum_{t=1}^{T} \Big( \mathrm{ucb}_t\big(\pi^\star(z_t), a_t^{-i}, z_t\big) - \mathrm{ucb}_t\big(a_t^i, a_t^{-i}, z_t\big) \Big) + \sum_{t=1}^{T} 2\beta_t \sigma_t\big(a_t^i, a_t^{-i}, z_t\big) \\
&\leq \underbrace{\sum_{t=1}^{T} \Big( \mathrm{ucb}_t\big(\pi^\star(z_t), a_t^{-i}, z_t\big) - \mathrm{ucb}_t\big(a_t^i, a_t^{-i}, z_t\big) \Big)}_{\hat{R}_c^i(T)} + 4\beta_T \sqrt{\gamma_T \lambda T}\,. 
\end{aligned}
\tag{11}
$$

The first inequality follows by the definition of $\mathrm{ucb}_t(\cdot)$ (see (3)), the specific choice of the confidence level $\beta_t$, and Lemma 7. The last inequality follows by [37, Lemma 5.4]. The rest of the proof proceeds to show that, with probability at least $1 - \delta/2$,

$$
\hat{R}_c^i(T) \leq \sqrt{T|\mathcal{Z}| \log K} + \sqrt{0.5T \log(2/\delta)}\,.
\tag{12}
$$

The theorem statement then follows by a standard union bound argument.

First, by straightforward application of the Hoeffding–Azuma inequality (e.g., [11, Lemma A.7]), it follows that with probability at least $1 - \delta/2$,

$$
\sum_{t=1}^{T} \underbrace{\Big| \mathrm{ucb}_t\big(a_t^i, a_t^{-i}, z_t\big) - \sum_{a \in \mathcal{A}^i} p_t(z_t)[a] \cdot \mathrm{ucb}_t\big(a, a_t^{-i}, z_t\big) \Big|}_{X_t} \leq \sqrt{0.5T \log(2/\delta)}\,,
\tag{13}
$$

since the variables $X_t$'s form a martingale difference sequence, being $\sum_{a \in \mathcal{A}^i} p_t(z_t)[a] \cdot \mathrm{ucb}_t\big(a, a_t^{-i}, z_t\big)$ the expected value of $\mathrm{ucb}_t\big(a_t^i, a_t^{-i}, z_t\big)$ conditioned on the history $\{a_\tau^i, a_\tau^{-i}, z_\tau, \epsilon_\tau\}_{\tau=1}^{t-1}$ and on context $z_t$. Then, using (13), $\hat{R}_c^i(T)$ can be bounded, with probability $1 - \delta/2$, as

$$
\begin{aligned}
\hat{R}_c^i(T) &\leq \sum_{t=1}^{T} \mathrm{ucb}_t\big(\pi^\star(z_t), a_t^{-i}, z_t\big) - \sum_{a \in \mathcal{A}^i} p_t(z_t)[a] \cdot \mathrm{ucb}_t\big(a, a_t^{-i}, z_t\big) + \sqrt{0.5T \log(2/\delta)} \\
&= \sum_{z \in \mathcal{Z}} \sum_{t:z_t=z} \mathrm{ucb}_t\big(\pi^\star(z_c), a_t^{-i}, z_c\big) - \sum_{a \in \mathcal{A}^i} p_t(z_c)[a] \cdot \mathrm{ucb}_t\big(a, a_t^{-i}, z_c\big) + \sqrt{0.5T \log(2/\delta)}\,.
\end{aligned}
\tag{14}
$$

At this point, we can use the properties of the MW rule used to compute the distribution $p_t(z) \in \Delta^K$. Note that, for each context $z \in \mathcal{Z}$, the distribution $p_t(z)$ computed by C.GP-MW precisely follows the MW rule (9) with the sequence of functions $\{\mathrm{ucb}_\tau(\cdot, a_\tau^{-i}, z)\}_{\tau:z_\tau=z}$ and the sequence of learning rates $\{\eta_\tau\}_{\tau=1}^{T_z} = \{2\sqrt{\log K/\tau}\}_{\tau=1}^{T_z}$, where $T_z = \sum_{\tau=1}^{t} \mathbb{1}\{z_\tau = z_t\}$ is the number of times context $z$ was revealed. Hence, we can apply Lemma 8 for each context $z$ and obtain:

$$
\begin{aligned}
\sum_{t:z_t=z} \mathrm{ucb}_t\big(\pi^\star(z_c), a_t^{-i}, z_c\big) - \sum_{a \in \mathcal{A}^i} & p_t(z_c)[a] \cdot \mathrm{ucb}_t\big(a, a_t^{-i}, z_c\big) \leq \\
&\leq 0.5\sqrt{T_z \log K} + \frac{2\sqrt{\log K}}{8} \sum_{\tau=1}^{T_z} \frac{1}{\sqrt{\tau}} \\
&\leq 0.5\sqrt{T_z \log K} + \frac{2\sqrt{\log K}}{8} 2\sqrt{T_z} = \sqrt{T_z \log K}\,.
\end{aligned}
\tag{15}
$$

Equation (15), together with the bound (14) leads to

$$\hat{R}_c^i(T) \leq \sum_{z \in \mathcal{Z}} \sqrt{T_z \log K} + \sqrt{0.5T \log(2/\delta)}$$
$$\leq \sqrt{T|\mathcal{Z}| \log K} + \sqrt{0.5T \log(2/\delta)},$$

where in the last inequality we have used Cauchy–Schwarz inequality and $\sum_{z \in \mathcal{Z}} T_z = T$. Hence, we finally proved (12). Therefore, with probability at least $1 - \delta/2 - \delta/2 = 1 - \delta$ we obtain the final regret bound combining (11) and (12). $\qquad \square$

## A.2  Proof of Theorem 1

*Proof.* Similarly to Appendix A.1, we let $\pi^\star = \arg\max_{\pi \in \Pi^i} \sum_{t=1}^T r(\pi(z_t), a_t^{-i}, z_t)$ and seek to bound $R_c^i(T) = \sum_{t=1}^T r(\pi^\star(z_t), a_t^{-i}, z_t) - r(a_t^i, a_t^{-i}, z_t)$ with high probability. Recall that Strategy 2 builds an $\epsilon$-net of the contexts space by creating new L1-balls in a greedy fashion. After $T$ game rounds, the set $\mathcal{C}$ contains the centers $z \in \mathcal{Z}$ of the balls created so far. Moreover, at each round $t$, the variable $z_t'$ indicates the ball that context $z_t$ has been associated to. According to this notation, player $i$'s regret can be rewritten as

$$R_c^i(T) = \sum_{z \in \mathcal{C}} \sum_{t:z_t'=z} r(\pi^\star(z_t), a_t^{-i}, z_t) - r(a_t^i, a_t^{-i}, z_t)$$
$$= \underbrace{\sum_{z \in \mathcal{C}} \sum_{t:z_t'=z} r(\pi^\star(z_t), a_t^{-i}, z_t) - r(\pi^\star(z), a_t^{-i}, z_t)}_{R_L(T)} + \underbrace{\sum_{z \in \mathcal{C}} \sum_{t:z_t'=z} r(\pi^\star(z), a_t^{-i}, z_t) - r(a_t^i, a_t^{-i}, z_t)}_{R_C(T)},$$

where in the last equality we have added and subtracted the term $\sum_{z \in \mathcal{C}} \sum_{t:z_t'=z} r(\pi^\star(z), a_t^{-i}, z_t)$.

The regret term $R_L(T)$ can be bounded using $L_r$-Lipschizness of $r(\cdot)$ in its first argument and $L_p$-Lipschizness of the optimal policy:

$$R_L(T) \leq \sum_{z \in \mathcal{C}} \sum_{t:z_t'=z} L_r \|\pi^\star(z_t) - \pi^\star(z)\|_1 \leq \sum_{z \in \mathcal{C}} \sum_{t:z_t'=z} L_r L_p \|z_t - z\|_1$$
$$\leq L_r L_p T \epsilon,$$

where in the last step we have used that, when $z_t' = z$, $z_t$ belongs to the ball centered at $z$.

We can now proceed in bounding $R_C(T)$. Note that we can apply the same proof steps of Appendix A.1 (namely, Equations (11) and (13)) to show that, with probability at least $1 - \delta$, we have:

$$R_C(T) \leq \sum_{z \in \mathcal{C}} \sum_{t:z_t'=z} \mathrm{ucb}_t(\pi^\star(z_t), a_t^{-i}, z_t) - \sum_{a \in \mathcal{A}^i} p_t(z_t)[a] \cdot \mathrm{ucb}_t(a, a_t^{-i}, z_t)$$
$$+ 4\beta_T \sqrt{\gamma_T \lambda T} + \sqrt{0.5T \log(2/\delta)}.$$

where we have conditioned on the event of Lemma 7 and applied the Hoeffding-Azuma inequality. At this point, we can use the properties of the MW rule used in Strategy 2 to compute $p_t(z_t)$ at each round. Note that Strategy 2 corresponds to maintaining a distribution for each $z \in \mathcal{C}$ and update it when context $z_t$ belongs to such ball. After $T$ rounds, for each $z \in \mathcal{Z}$ let $T_z = \sum_{t=1}^T \mathbb{1}\{z_t' = z\}$ be the number of times the revealed context belonged to the ball centered at $z$. Hence, for each $z \in \mathcal{C}$, a straightforward application of Lemma 8 leads to:

$$\sum_{t:z_t'=z} \mathrm{ucb}_t(\pi^\star(z_t), a_t^{-i}, z_t) - \sum_{a \in \mathcal{A}^i} p_t(z_t)[a] \cdot \mathrm{ucb}_t(a, a_t^{-i}, z_t) \leq$$
$$\leq 0.5\sqrt{\log K T_z} + \frac{2\sqrt{\log K}}{8} \sum_{\tau=1}^{T_z} \frac{1}{\sqrt{\tau}}$$

$$\leq 0.5\sqrt{\log K T_z} + \frac{2\sqrt{\log K}}{8}2\sqrt{T_z} = \sqrt{T_z \log K}\,.$$

where we have used the same steps to obtain Equation (15) in Appendix A.1. Hence, summing over all the contexts in $\mathcal{C}$ we obtain

$$
\begin{aligned}
R_C(T) &\leq \sum_{z \in \mathcal{C}} \sqrt{T_c \log K} \quad + 4\beta_T\sqrt{\gamma_T \lambda T} + \sqrt{0.5T \log(1/\delta_2)} \\
&\leq \sqrt{T|\mathcal{C}| \log K} \quad + 4\beta_T\sqrt{\gamma_T \lambda T} + \sqrt{0.5T \log(1/\delta_2)} \\
&= \epsilon^{-c/2}\sqrt{T \log K} \quad + 4\beta_T\sqrt{\gamma_T \lambda T} + \sqrt{0.5T \log(1/\delta_2)}\,.
\end{aligned}
$$

In the last inequality we have used $|\mathcal{C}| \leq (1/\epsilon)^c$, because the contexts space $\mathcal{Z} \subseteq [0,1]^c$ can be covered by at most $(1/\epsilon)^c$ balls of radius $\epsilon$ such that the distance between their centers is at least $\epsilon$.

Therefore, combining the bounds for $R_L(T)$ and $R_C(T)$, the contextual regret of player $i$ is bounded, with probability at least $1 - \delta$, by:

$$
\begin{aligned}
R_c^i(T) &\leq L_r L_p T\epsilon + \epsilon^{-c/2}\sqrt{T \log K} + 4\beta_T\sqrt{\gamma_T \lambda T} + \sqrt{0.5T \log(2/\delta)} \\
&= (L_r L_p)^{\frac{c}{c+2}} T^{\frac{c+1}{c+2}} + (L_r L_p)^{\frac{c}{c+2}} T^{\frac{c+1}{c+2}}\sqrt{\log K} + 4\beta_T\sqrt{\gamma_T \lambda T} + \sqrt{0.5T \log(2/\delta)}\,.
\end{aligned}
$$

where we have substituted the choice of $\epsilon = (L_r L_p)^{-\frac{2}{c+2}} T^{-\frac{1}{c+2}}$. $\qquad\square$

### A.3  Proof of Theorem 2

*Proof.* We let $R_c^i(\pi, T) = \sum_{t=1}^T r\big(\pi(z_t), a_t^{-i}, z_t\big) - r\big(a_t^i, a_t^{-i}, z_t\big)$ be the regret of player $i$ with respect to a generic policy $\pi : \mathcal{Z} \to \mathcal{A}^i$. Our goal is to bound $R_c^i(\pi, T)$ in expectation, with respect to the random sequence of contexts and played actions. For ease of exposition, we use the notation $v_{\ldots T}$ to indicate the sequence of variables $v_1, \ldots, v_T$. Moreover, we will explicitly consider an adaptive adversary that selects $a_t^{-i}$ as a function $a_t^{-i} = f(\mathcal{H}_{t-1})$ of the history $\mathcal{H}_{t-1} := \{a_{\ldots t-1}^i, z_{\ldots t-1}\}$ but not of $z_t$.

First, note that the expected value of $R_c^i(\pi, T)$ is still a random variable which depends on the realization of the observation noise $\epsilon_t$'s. As it was done in proof of Theorems 1 and Appendix A.1, we can condition on the event of the confidence Lemma 7, and state that, with probability at least $1 - \delta/2$, it can be bounded by

$$
\begin{aligned}
\mathbb{E}_{\substack{z_{\ldots T} \\ a_{\ldots T}^i}}\big[R_c^i(\pi, T)\big] &= \mathbb{E}_{\substack{z_{\ldots T} \\ a_{\ldots T}^i}}\left[\sum_{t=1}^T r\big(\pi(z_t), a_t^{-i}, z_t\big) - r\big(a_t^i, a_t^{-i}, z_t\big)\right] \\
&\leq \mathbb{E}_{\substack{z_{\ldots T} \\ a_{\ldots T}^i}}\left[\sum_{t=1}^T \mathrm{ucb}_t\big(\pi(z_t), a_t^{-i}, z_t\big) - \mathrm{ucb}_t\big(a_t^i, a_t^{-i}, z_t\big) + \sum_{t=1}^T 2\beta_t \sigma_t\big(a_t^i, a_t^{-i}, z_t\big)\right] \\
&\leq \mathbb{E}_{\substack{z_{\ldots T} \\ a_{\ldots T}^i}}\underbrace{\left[\sum_{t=1}^T \mathrm{ucb}_t\big(\pi(z_t), a_t^{-i}, z_t\big) - \mathrm{ucb}_t\big(a_t^i, a_t^{-i}, z_t\big)\right]}_{\hat{R}_c^i(\pi, T)} + 4\beta_T\sqrt{\gamma_T \lambda T}, \quad (16)
\end{aligned}
$$

where we have used the definition of $\mathrm{ucb}_t(\cdot)$, $\beta_t$ set according to Lemma 7, and [37, Lemma 5.4].

Now, we consider a generic sequence $\{\epsilon_t\}_{t=1}^T$ of noise realizations and proceed bounding the expected value of $\hat{R}_c^i(\pi, T)$ for any of such sequences. Moreover, as in [28] we will make use of a *ghost sample* $z_0 \sim \zeta$ which is sampled from the contexts' distribution independently from the whole history $\mathcal{H}_T$ of the game. Also, we now explicitly consider the adaptiveness of the adversary. Using the law of total expectation, the expected value of $\hat{R}_c^i(\pi, T)$ can be rewritten as

$$
\mathbb{E}_{\substack{z_{\ldots T} \\ a_{\ldots T}^i}}\big[\hat{R}_c^i(\pi, T)\big] = \mathbb{E}_{\substack{z_{\ldots T} \\ a_{\ldots T}^i}}\left[\sum_{t=1}^T \mathrm{ucb}_t\big(\pi(z_t), f(\mathcal{H}_{t-1}), z_t\big) - \mathrm{ucb}_t\big(a_t^i, f(\mathcal{H}_{t-1}), z_t\big)\right]
$$

$$= \mathop{\mathbb{E}}_{\substack{z_{\dots T} \\ a^i_{\dots T}}} \left[ \sum_{t=1}^T \mathbb{E}_{z_t, a^i_t} \big[ \mathrm{ucb}_t\big(\pi(z_t), f(\mathcal{H}_{t-1}), z_t\big) - \mathrm{ucb}_t\big(a^i_t, f(\mathcal{H}_{t-1}), z_t\big) \mid \mathcal{H}_{t-1} \big] \right]$$

$$= \mathop{\mathbb{E}}_{\substack{z_{\dots T} \\ a^i_{\dots T}}} \left[ \sum_{t=1}^T \mathbb{E}_{z_t} \big[ \mathrm{ucb}_t\big(\pi(z_t), f(\mathcal{H}_{t-1}), z_t\big) - \sum_{a \in \mathcal{A}^i} p_t(z_t)[a] \cdot \mathrm{ucb}_t\big(a, f(\mathcal{H}_{t-1}), z_t\big) \mid \mathcal{H}_{t-1} \big] \right]$$

$$= \mathop{\mathbb{E}}_{\substack{z_{\dots T} \\ a^i_{\dots T}}} \left[ \sum_{t=1}^T \mathbb{E}_{z_0} \big[ \mathrm{ucb}_t\big(\pi(z_0), f(\mathcal{H}_{t-1}), z_0\big) - \sum_{a \in \mathcal{A}^i} p_t(z_0)[a] \cdot \mathrm{ucb}_t\big(a, f(\mathcal{H}_{t-1}), z_0\big) \mid \mathcal{H}_{t-1} \big] \right]$$

$$= \mathop{\mathbb{E}}_{\substack{z_{\dots T} \\ a^i_{\dots T} \\ z_0}} \left[ \sum_{t=1}^T \mathrm{ucb}_t\big(\pi(z_0), f(\mathcal{H}_{t-1}), z_0\big) - \sum_{a \in \mathcal{A}^i} p_t(z_0)[a] \cdot \mathrm{ucb}_t\big(a, f(\mathcal{H}_{t-1}), z_0\big) \right] \qquad (17)$$

The second equality follows by the law of total expectation. The third equality holds since, conditioned on the history $\mathcal{H}_{t-1}$, $a^i_t$ is distributed according to $p_t(z_t)$. The fourth equality follows since $z_t$ and $z_0$ have the same distribution and the functions $\mathrm{ucb}_t(\cdot)$ and $p_t(\cdot)$ do not depend on the realization of $z_t$. The last inequality is obtained by applying again the law of total expectation.

At this point, we can apply Lemma 8 considering the sequence of functions $g_1, \dots, g_T$ with $g_\tau(\cdot) = \mathrm{ucb}_\tau(\cdot, f(\mathcal{H}_{\tau-1}), z_0)$ for $\tau = 1, \dots, T$ and noting that, for each $z_0$, $p_t(z_0)$ computed using the MW rule (5) corresponds to the distribution computed according to rule (9) for each $t = 1, \dots, T$. Therefore, (17) implies that

$$\mathop{\mathbb{E}}_{\substack{z_{\dots T} \\ a^i_{\dots T}}} \big[ \hat{R}^i_c(\pi, T) \big] \leq \frac{\log K}{\eta_T} + \frac{\sum_{t=1}^T \eta_t}{8} \,.$$

Finally, the theorem statement is obtained substituting the bound above in (16) and considering the constant learning rate $\eta_t = \sqrt{8 \log(K)/T}$. $\qquad \square$

# B  Supplementary Material for Section 4

## B.1  Proof of Proposition 4 (Finite-time approximation of c-CCEs)

*Proof.* After $T$ rounds of the contextual game, consider a generic player $i$. By definition of contextual regret, see (1), we have

$$\frac{1}{T} \sum_{t=1}^T r^i(a_t, a_t^{-i}, z_t) \geq \frac{1}{T} \sum_{t=1}^T r^i(\pi(z_t), a_t^{-i}, z_t) - \frac{R^i_c(T)}{T} \qquad \forall \pi \in \Pi^i \,. \qquad (18)$$

Let now $\rho_T$ be the empirical policy up to time $T$, defined as in Section 4.1. Then, it is not hard to verify that the above cumulative rewards for player $i$ can be written as

$$\frac{1}{T} \sum_{t=1}^T r^i(a_t, a_t^{-i}, z_t) = \frac{1}{T} \sum_{t=1}^T \mathop{\mathbb{E}}_{\mathbf{a} \sim \rho_T(z_t)} r(\mathbf{a}, z_t) \,,$$

$$\frac{1}{T} \sum_{t=1}^T r^i(\pi(z), a_t^{-i}, z_t) = \frac{1}{T} \sum_{t=1}^T \mathop{\mathbb{E}}_{\mathbf{a} \sim \rho_T(z_t)} r(\pi(z), a^{-i}, z_t) \,.$$

Therefore, (18) becomes:

$$\frac{1}{T} \sum_{t=1}^T \mathop{\mathbb{E}}_{\mathbf{a} \sim \rho_T(z_t)} r(\mathbf{a}, z_t) \geq \frac{1}{T} \sum_{t=1}^T \mathop{\mathbb{E}}_{\mathbf{a} \sim \rho_T(z_t)} r(\pi(z), a^{-i}, z_t) - \frac{R^i_c(T)}{T} \qquad \forall \pi \in \Pi^i \,. \qquad (19)$$

Note that this is precisely the condition of $\epsilon$-c-CCE (see Definition 3) for player $i$. The final result is then simply obtained by considering the player with the highest regret. $\qquad \square$

## B.2 An alternative notion of c-CCE for stochastic contexts

In Section 4 we defined the notion of c-CCE (Definition 3) for a contextual game described by an arbitrary sequence of contexts $z_1, \ldots, z_T$. In this section, we consider the case in which contexts are stochastic samples from the same distribution $\zeta$, i.e., $z_t \sim \zeta$ for all $t$. In such a case, the following alternative notion of c-CCE can be defined by considering the *expected* context realization (rather than considering the time-averaged game as in Definition 3).

**Definition 10.** *Consider a contextual game and assume contexts are sampled i.i.d. from distribution $\zeta$. A contextual coarse-correlated equilibrium for stochastic contexts (c-$\zeta$-CCE) is a policy $\rho : \mathcal{Z} \to \Delta^{|\mathcal{A}|}$ mapping contexts to distributions over $\mathcal{A}$ such that:*

$$\mathbb{E}_{z \sim \zeta} \mathbb{E}_{\mathbf{a} \sim \rho(z)} r^i(\mathbf{a}, z) \geq \mathbb{E}_{z \sim \zeta} \mathbb{E}_{\mathbf{a} \sim \rho(z)} r^i(\pi(z), a^{-i}, z) \quad \forall \pi \in \Pi^i, \quad \forall i = 1, \ldots, N. \tag{20}$$

*Moreover, $\rho$ is an $\epsilon$-c-$\zeta$-CCE if the above inequality is satisfied up to an $\epsilon \in \mathbb{R}_+$ accuracy.*

Similarly to Proposition 4, the following proposition shows that, in this specific setting, c-$\zeta$-CCEs can also be approached whenever players minimize their contextual regrets.

**Proposition 11** (Asymptotic and finite-time convergence to c-$\zeta$-CCEs). *Consider a contextual game and assume contexts are sampled i.i.d. from distribution $\zeta$. Let $\rho_T$ be the empirical policy at round $T$. Then, as $T \to \infty$, if players have vanishing contextual regrets, $\rho_T$ converges to a c-$\zeta$-CCE almost surely. Moreover, after $T$ game rounds, let $R_c^i(T)$'s denote the players' contextual regrets, $\delta \in (0, 1)$, and assume $\mathcal{Z}$ is finite. Then, with probability at least $1 - \delta$, $\rho_T$ is a $\epsilon$-c-$\zeta$-CCE with*

$$\epsilon \leq 2 \sqrt{\frac{\log(|\mathcal{Z}| \cdot |\mathcal{A}|)}{2} + \frac{\log(2/\delta)}{2T}} + \max_{i \in \{1, \ldots, N\}} \frac{R_c^i(T)}{T}.$$

Compared to CCEs (and c-CCEs), c-$\zeta$-CCEs can be approximated in finite time only with high-probability and with an extra approximation factor of $\mathcal{O}(\log(|\mathcal{Z}| |\mathcal{A}|) + \log(1/\delta)/T)$. Intuitively, this is because the empirical distribution of observed contexts needs to concentrate around the true contexts distribution $\zeta$. We recover asymptotic convergence to c-$\zeta$-CCEs since such distribution converges to $\zeta$ with probability 1.

*Proof.* By definition of contextual regret, see (1), for each player $i$

$$\frac{1}{T} \sum_{t=1}^T r^i(a_t, a_t^{-i}, z_t) \geq \frac{1}{T} \sum_{t=1}^T r^i(\pi(z_t), a_t^{-i}, z_t) - \frac{R_c^i(T)}{T} \qquad \forall \pi \in \Pi^i. \tag{21}$$

Let $\zeta_T$ be the empirical distribution of observed contexts. Moreover, let $\rho_T$ be the empirical policy up to time $T$, defined in Section 4.1. Then, following the same steps of Proof of Proposition 4:

$$\frac{1}{T} \sum_{t=1}^T r^i(a_t, a_t^{-i}, z_t) = \mathbb{E}_{z \sim \zeta_T} \mathbb{E}_{\mathbf{a} \sim \rho_T(z)} r(\mathbf{a}, z),$$

$$\frac{1}{T} \sum_{t=1}^T r^i(\pi(z), a_t^{-i}, z_t) = \mathbb{E}_{z \sim \zeta_T} \mathbb{E}_{\mathbf{a} \sim \rho_T(z)} r(\pi(z), a^{-i}, z).$$

Therefore, (21) rewrites as

$$\mathbb{E}_{z \sim \zeta_T} \mathbb{E}_{\mathbf{a} \sim \rho_T(z)} r(\mathbf{a}, z) \geq \mathbb{E}_{z \sim \zeta_T} \mathbb{E}_{\mathbf{a} \sim \rho_T(z)} r(\pi(z), a^{-i}, z) - \frac{R_c^i(T)}{T} \qquad \forall \pi \in \Pi^i. \tag{22}$$

As $T \to \infty$, $\zeta_T \to \zeta$ as contexts are i.i.d. samples from $\zeta$. Moreover, if players use no-regret strategies, $R_c^i(T)/T \to 0$ for $i = 1, \ldots N$ and hence the above inequality implies that $\rho_T$ converges to a c-$\zeta$-CCE (see Definition 10).

For finite $T$, the above inequality resembles the desired c-$\zeta$-CCE condition, with the difference that the outer expectations are taken with respect to the empirical contexts' distribution $\zeta_T$ instead of the true one. To cope with this, we show that such expectations indeed concentrate, up to some accuracy,

around the expectations with respect to the true distribution $\zeta$. More precisely, we show that with probability at least $1 - \delta$

$$\left| \underset{z \sim \zeta_T}{\mathbb{E}} f(z, \rho_T) - \underset{z \sim \zeta}{\mathbb{E}} f(z, \rho_T) \right| \leq \sqrt{\frac{\log(|\mathcal{Z}| \cdot |\mathcal{A}|)}{2} + \frac{\log(2/\delta)}{2T}}, \qquad (23)$$

where $f(z, \rho_T) = \mathbb{E}_{\mathbf{a} \sim \rho_T(z)} r(\mathbf{a}, z)$. Moreover, the same condition holds for $f(z, \rho_T) = \mathbb{E}_{\mathbf{a} \sim \rho_T(z)} r(\pi(z), a^{-i}, z)$ for each $\pi \in \Pi^i$. Combined with (22), this implies that for each player $i$ and each $\pi \in \Pi^i$, with probability $1 - \delta$,

$$\underset{z \sim \zeta}{\mathbb{E}} \underset{\mathbf{a} \sim \rho_T(z)}{\mathbb{E}} r(\mathbf{a}, z) \geq \underset{z \sim \zeta}{\mathbb{E}} \underset{\mathbf{a} \sim \rho_T(z)}{\mathbb{E}} r(\pi(z), a^{-i}, z) - 2 \sqrt{\frac{\log(|\mathcal{Z}| \cdot |\mathcal{A}|)}{2} + \frac{\log(2/\delta)}{2T}} - \frac{R_c^i(T)}{T},$$

which would prove Proposition 11.

It remains to show (23). For a *given* policy $\rho : \mathcal{Z} \to \Delta^{|\mathcal{A}|}$, a straightforward application of Hoeffding's inequality [21] shows that for any $\epsilon > 0$

$$\mathbb{P} \left[ \left| \underset{z \sim \zeta_T}{\mathbb{E}} f(z, \rho) - \underset{z \sim \zeta}{\mathbb{E}} f(z, \rho) \right| > \epsilon \right] \leq 2 \exp\left( -2T\epsilon^2 \right), \qquad (24)$$

where we have used the fact that $f(z, \rho) = \mathbb{E}_{\mathbf{a} \sim \rho(z)} r^i(\mathbf{a}, z) \in [0, 1]$ and that $z_1, \ldots, z_T$ are i.i.d. sampled from $\zeta$. Unfortunately, we cannot apply the condition above directly to the empirical policy $\rho_T$, since it is not fixed a-priori, but is computed as a function of the realized samples $z_1, \ldots, z_T$. However, we can consider the set $\mathcal{P}_T$ of all the possible empirical policies $\rho : \mathcal{Z} \to \Delta^{|\mathcal{A}|}$ resulting from $T$ rounds of the repeated game. Note that each of such policies is uniquely defined by the sequence $\{a_t^i, a_t^{-i} z_t\}_{t=1}^T$ of revealed contexts and actions played up to round $T$. Therefore, $\mathcal{P}_T$ is a finite set of cardinality $|\mathcal{P}_T| = (|\mathcal{Z}| \cdot |\mathcal{A}|)^T$. Hence, it holds

$$\mathbb{P} \left[ \left| \underset{z \sim \zeta_T}{\mathbb{E}} f(z, \rho_T) - \underset{z \sim \zeta}{\mathbb{E}} f(z, \rho_T) \right| > \epsilon \right] \leq \mathbb{P} \left[ \sup_{\rho \in \mathcal{P}_T} \left| \underset{z \sim \zeta_T}{\mathbb{E}} f(z, \rho) - \underset{z \sim \zeta}{\mathbb{E}} f(z, \rho) \right| > \epsilon \right]$$

$$= \mathbb{P} \left[ \bigcup_{\rho \in \mathcal{P}_T} \left\{ \left| \underset{z \sim \zeta_T}{\mathbb{E}} f(z, \rho) - \underset{z \sim \zeta}{\mathbb{E}} f(z, \rho) \right| > \epsilon \right\} \right]$$

$$\leq |\mathcal{P}_T| \, \mathbb{P} \left[ \left| \underset{z \sim \zeta_T}{\mathbb{E}} f(z, \rho) - \underset{z \sim \zeta}{\mathbb{E}} f(z, \rho) \right| > \epsilon \right]$$

$$\leq 2 |\mathcal{P}_T| \, \exp\left( -2T\epsilon^2 \right).$$

The first equality holds since, given a set of random variables $x_1, \ldots, x_n$, asking that $\sup_i x_i > \epsilon$ is equivalent to asking that at least one of the $x_i$'s is greater than $\epsilon$. The second inequality is a standard probability union bound, while the last inequality follows by (24). This proves (23) after setting the right hand side equal to $\delta$ and substituting $|\mathcal{P}_T| = (|\mathcal{Z}| \cdot |\mathcal{A}|)^T$.

$\square$

### B.3 Proof of Proposition 6 (Convergence to approximate efficiency)

*Proof.* For ease of notation, let $\pi_\star^1, \ldots, \pi_\star^N$ be the optimal policies that solve (8), so that $\mathrm{OPT} = \frac{1}{T} \sum_{t=1}^T \Gamma\left( \pi_\star^1(z_t), \ldots, \pi_\star^N(z_t), z_t \right)$. Using the definitions of contextual regret and $(\lambda, \mu)$-smoothness, the sum of cumulative rewards can be lower bounded as:

$$\frac{1}{T} \sum_{t=1}^T \sum_{i=1}^N r^i(a_t^i, a_t^{-i}, z_t)$$

$$\geq \frac{1}{T} \sum_{t=1}^T \sum_{i=1}^N r^i\left( \pi_\star^i(z_t), a_t^{-i}, z_t \right) - \sum_{i=1}^N \frac{R_c^i(T)}{T}$$

$$\geq \frac{1}{T}\sum_{t=1}^{T}\Big[\lambda(z_t)\cdot\Gamma\big(\pi_\star^1(z_t),\ldots,\pi_\star^N(z_t),z_t\big)-\mu(z_t)\cdot\Gamma(a_t^1,\ldots,a_t^N,z_t)\Big]\quad-\sum_{i=1}^{N}\frac{R_c^i(T)}{T}$$

$$\geq \bar\lambda\cdot\mathrm{OPT}-\bar\mu\cdot\frac{1}{T}\sum_{t=1}^{T}\Gamma(a_t^1,\ldots,a_t^N,z_t)\quad-\sum_{i=1}^{N}\frac{R_c^i(T)}{T}\,.$$

In the first inequality we have used the definition of contextual regret (see (1)) with respect to policy $\pi_\star^i$ for each player (note that $\pi_\star^i$ is not necessarily the optimal policy in hindsight for player $i$). In the second inequality we have used the fact that the game is $\big(\lambda(z_t),\mu(z_t)\big)$-smooth at each time $t$ and applied condition (7) with outcomes $\mathbf{a}_1=(a_t^1,\ldots,a_t^N)$ and $\mathbf{a}_2=\big(\pi_\star^1(z_t),\ldots,\pi_\star^N(z_t)\big)$. The last inequality follows from the definition of $\bar\lambda,\bar\mu$, and OPT.

At this point, note that $\frac{1}{T}\sum_{t=1}^{T}\Gamma(a_t^1,\ldots,a_t^N,z_t)\geq\frac{1}{T}\sum_{t=1}^{T}\sum_{i=1}^{N}r^i(a_t^i,a_t^{-i},z_t)$ since by definition of social welfare $\Gamma(\mathbf{a},z)\geq\sum_{i=1}^{N}r^i(\mathbf{a},z)$ for all $(\mathbf{a},z)$. Then, the above inequalities imply that

$$\frac{1}{T}\sum_{t=1}^{T}\Gamma(a_t^1,\ldots,a_t^N,z_t)\geq\bar\lambda\cdot\mathrm{OPT}-\bar\mu\cdot\frac{1}{T}\sum_{t=1}^{T}\Gamma(a_t^1,\ldots,a_t^N,z_t)\quad-\sum_{i=1}^{N}\frac{R_c^i(T)}{T}\,,$$

which after rearranging yields the desired result. $\qquad\square$

## C    Contextual Traffic Routing - Experimental Setup

In this section we describe the experimental setup of the contextual traffic routing game of Section 5. We consider the traffic network of Sioux-Falls, a directed graph with 24 nodes and 76 edges and use the game model of [32]. Data from [25, 1] include node coordinates and capacities $C_e\in\mathbb{R}_+$ of each network's edge $e=1,\ldots,76$. Moreover, data also include the units (e.g., cars) that need to be sent from any node to any other node in the network, for a total of 528 distinct origin-destination pairs. Hence, we let $N=528$ be the number of agents in the network and assume, at every round, each agent $i$ needs to send $d^i$ units from origin node $O^i$ to destination node $D^i$. In order to send these units, each agent can choose one of the $K=5$ shortest routes between $O^i$ and $D^i$. We let $x_t^i\in\mathbb{R}^{76}$ represent the route chosen by agent $i$ at round $t$, where $x_t^i[e]=d_i$ if edge $e$ belongs to such route, and $x_t^i[e]=0$ otherwise. Moreover, we let $x_t^{-i}=\sum_{j\neq i}x_t^j$ represent the routes chosen by the rest of the agents.

At each round, the network displays different capacities (network's capacities represent the contextual information of the game) which are observed by the agents and should be used to choose better routes, depending on the circumstances. This is different from the game model of [32] where network capacities are assumed constant. We let the context vector $z_t\in\mathbb{R}_+^{76}$ represent the network's capacities at round $t$, and assume each $z_t$ is i.i.d. sampled from a static distribution $\zeta$. The contexts distribution $\zeta$ is generated as follows. We let $\mathcal{Z}$ be a set of 10 randomly generated capacity profiles $z$ where $z[e]$ is uniformly distributed in $[0,1.2\cdot C_e]$ for $e=1,\ldots,76$. Then, we let $\zeta$ be the uniform distribution over $\mathcal{Z}$.

Given context $z_t$ and routes $x_t^i$, $x_t^{-i}$, the reward of each agent $i$ is:

$$r^i(x_t^i,x_t^{-i},z_t)=-\sum_{e=1}^{76}x_t^i[e]\cdot t_e(x_t^i+x_t^{-i},z_t[e])\,,$$

where $t_e(\cdot)$ is the traveltime function of edge $e$ (i.e., the relation between number of units traversing edge $e$ and the time needed to traverse it). Such functions are unknown to the agents, and according to [25, 1] are defined by the Bureau of Public Roads (BPR) congestion model:

$$t_e(x,z)=f_e\cdot\left(1+0.15\Big(\frac{x}{z}\Big)^4\right),$$

where $f_e\in\mathbb{R}_+$ is the free-flow traveltime of edge $e$ (also provided by the network's data). At the end of each round, hence, we quantify the congestion of each edge $e$ with the quantity $0.15((x_t^i[e]+x_t^{-i}[e])/z_t[e])^4$.

To run our experiments, we estimate upper and lower bounds on the agents' rewards by sampling $10'000$ random contexts and game outcomes, and feed such bounds to the agents so that rewards can be scaled in the $[0, 1]$ interval. Moreover, at each round agents receive a noisy measurement of their rewards, with noise standard deviation $\sigma$ set to $0.1\%$. To run ROBUSTLINEXP3 [28, Theorem 1] we set learning rate $\eta = 0.3$ and exploration parameter $\gamma = 0.2$ (we observe worse performance when setting them to their theoretical values). For GP-MW we use the composite kernel $k(x_t^i, x_t^{-i}, z_t) = k_1(x_t^i) * k_2(x_t^i + x_t^{-i})$ used also in [32], while for C.GP-MW the kernel $k(x_t^i, x_t^{-i}, z_t) = k_1(x_t^i) * k_2((x_t^i + x_t^{-i})/z_t)$, where $k_1$ is a linear kernel and $k_2$ is a polynomial kernel of degree 4. However, we observe similar performance when polynomials of different degrees are used or when $k_2$ is the widely used SE kernel. Kernel hyperparameters are optimized offline over 100 random datapoints and kept fixed. We set $\eta_t$ according to Theorems 1 and 2, and confidence level $\beta_t = 2.0$ (theoretical values for $\beta_t$ are found to be overly conservative, as also observed in [37, 32]).