[Reviews · NeurIPS 2020]

Review 1

Summary and Contributions: This paper introduces the notion of contextual games, a class of games where the payoff of each player depends on both the action profile, and the current context. A typical example of such games is the routing games, where the delay faced by an agent not only depends on the paths chosen by all the agents, but also on the context e.g. capacity of the edges, weather etc. From a single agent’s point of view, this is just an adversarial contextual bandit problem and the existing algorithms can be used to get a no-regret learning algorithm. However, the paper imposes kernel-based regularity assumptions to get improved bounds on the regret. In particular, the main contributions are the following: 1. When the number of contexts is a small finite set Z the authors prove a bound of \sqrt{T|Z| \log K} (*) 2. When the contexts are chosen from a c-dimensional space, the authors prove a bound of \sqrt{c TK\log K} 3. Then the authors introduce the concept of contextual-coarse correlated equilibrium and show that if the players follow the proposed contextual bandit algorithms, then each player gets vanishing regret. Moreover an extension of the standard price of anarchy argument shows that the optimal contextual welfare can be achieved unto the PoA ratio. (*) * Please check my comments later regarding these points.

Strengths: 1. The main contribution of the paper is in introducing the idea of contextual games. These games are more general than standard repeated games, and are better models of routing games. On the other hand, they are not hard to solve like stochastic games. 2. Another contribution is to use kernel-based regularity assumptions to derive improved bounds for contextual bandits. However, as I point out in the Relation to prior work section, this is not entirely new. 3. I view this paper mainly making a new theoretical contribution to the literature of multi-armed bandits and learning on games and it is definitely of interest to the NeurIPS community. 4. I think the theorems and the proofs are correct. However, I am not convinced by the claim that the dependence on K can be improved to be logarithmic in K, as the paper provides no bound on the term \gamma_T.

Weaknesses: 1. One of the terms appearing in regret (theorem 1 and 2) is the term \sqrt{\gamma_T T}. But how does the term \gamma_T change as we increase number of actions K, and number of agents N? For an upper bound, the authors provide two examples — d log T and log^{d+1} T where d is the dimension of the space consisting of action profile and contexts as tuples. Since the action tuples grow exponentially with N, could it be that d becomes large very soon and dominates the other term in the regret. It seems to me that depending on the dimension of the domain, the claim that regret depends only logarithmically on number of actions K, might not hold. 2. The authors rightly point out that from a single agent’s point of view, the problem can be thought of as an adversarial contextual bandits. However, instead of using existing algorithms they try to design new algorithms to get improved bounds. But what is the limit of such improvements if we just make some smoothness assumptions. The paper did not prove or discuss any lower bounds. 3. The price of anarchy result (proposition 6) effectively replaces the sequence of lambda-s and mu-s by their maximum and minimum respectively. But I don’t think this is the right generalization of the classic result for the contextual setting. Why should the price of anarchy should be bad if there is just one round t that gives poor choices of lambda(z_t) and mu(z_t)?

Correctness: I read the paper carefully and I think the theorems and propositions are correct as they are stated. I am not entirely convinced that the regret scales logarithmically in K, as the authors say in line 63. What if the term \gamma_t grows faster than log K and dominates the first term in the regret? I thought the experiment was correct and definitely showed that the proposed method dominates the other approaches.

Clarity: I thought the paper was well-written and I did not have any major problem following the arguments presented in the paper. However, I have a few suggestions. > I would have liked to see some more details about the GP-MW algorithm since it is crucial for designing algorithm 1. > It might be a good idea to move the background on price of anarchy to the related work section. > The term pseudo-regret was used without a definition. > The main text introduced the term \gamma_T and said that it is sample-complexity parameter that can be bounded analytically. Perhaps, in the introduction, the authors should highlight why such a term is necessary, and provide more explanation of its definition.

Relation to Prior Work: The authors claim that prior literature on contextual bandits do not exploit the fact that similar contexts and outcomes should produce similar rewards. However, there have been some attempts to model such similarity information. For example, > Contextual Bandits with Similarity Information, Slivkins, 2011. >Showing Relevant Ads via Lipschitz Context Multi-Armed Bandits, Lu et. al. 2010. Both the papers impose that reward when considered as a function of context and action is Lipschitz continuous. Lu et. al. also derives a regret bound of the form T^{(c+1)/(c+2)} which appears as one of the terms in theorem 1. It will be helpful if the authors could clarify how the kernel-based regularity assumption compares with these approaches. The authors should have also discussed the results on data-dependent type bounds for contextual bandits. The authors claimed that their regret bound scales logarithmically with K (number of actions), compared to sort{K} bound for the previous algorithms for contextual bandits. Recently, the following paper gave some improvement in this regard: > Taking a Hint: How to Leverage Loss Predictors in Contextual Bandits, Wei et. al. 2020. Otherwise, I felt that the paper covered the prior literature carefully.

Reproducibility: Yes

Additional Feedback: Update after Rebuttal: The authors addressed my criticisms about the dependence on \gamma_T and the price of anarchy. However, in the worst-case, \gamma_T can increase linearly with N and this implies that the proposed bound is meaningful in a setting with a small number of agents. Additionally, the authors acknowledged my comment about the price of anarchy but did not provide a way to improve the bound. Therefore, I am keeping my score as it is.


Review 2

Summary and Contributions: The paper introduces contextual games, which are repeated games such that at each turn there is a context all players can see, that effects their reward functions. It is assumed that the reward functions are smooth with respect to some kernel, such that the rewards from similar contexts are similar. Players have unknown reward functions, but they can observe the strategy profile at the end of each turn. The authors propose a multiplicative-weight type algorithm that achieves sublinear regret when played against other players and an oblivious adversary that chooses the context sequence. Based on this result, it is shown that the dynamics of the game converge to a contextual-coarse-correlated equilibrium (introduced and defined here). For smooth games, a bound on the gap from the social optimum is provided. Enhanced result for the case of i.i.d. contexts are also shown. The proposed algorithm is then demonstrated on a traffic routing game.

Strengths: This is a nice and creative paper that provides a framework that can be useful for the ML community. The theoretical results are sound and interesting, and the experiments serve as an interesting use case.

Weaknesses: From a technical point of view, the result is an incremental enhancement of [32], and follows by connecting known results. As such, the significance of the paper relies on the originality and usefulness of the novel framework of contextual games. This is by itself of course fine, since impact and usefulness are possibly the most important aspects anyway. The main weakness of this paper is that the usefulness and motivation of the results are a bit vague. The reason is that it's not clear why would selfish players follow the proposed algorithm. From a game-theoretic point of view, the approach here misses the dynamic aspect of repeated games. A more meaningful equilibrium notion for a repeated game takes into account the known history of the game (i.e., subprefect equilibrium). From an online learning point of view, the standard story that "bypasses" this conceptual difficulty is that each agent doesn't know it's facing other agents, and plays a good (no-regret) algorithm to guarantee its performance against any arbitrary adversary. If this adversary happens to be N-1 other players, then the nice finding is that a Nash equilibrium emerges from the interaction. However, this paper assumes that players can observe the strategy profile. Even more, the algorithm requires players to approximate the reward functions. With approximately known reward functions that are fully aware of the presence of N-1 other players, it's not clear why would players agree to run Algorithm 1 while they can adopt dynamic sophisticated strategies (i.e., Folk Theorems). For this reason, I find Lines 76-77 a bit misleading (what important results are recovered? better to be explicit). The (somewhat implicit) assumption that T is known also doesn't contribute to see in which kind of scenario running Algorithm 1 makes sense. Of course, players don't have to be selfish. But if players are cooperative, then it's not clear why would they want to converge to a contextual-coarse-correlated-equilibrium. The smoothness assumption along with the provided bound for the gap from social optimum do provide some justification as to why would one wants to consider this algorithm from a cooperative point of view. The experiments also support this. However, if this is the main motivation, it should be discussed and compared to more cooperative approaches. My hunch is that this is the best way to strengthen the contribution of this paper. However, I tend to like the paper despite these weaknesses. I'm very open to hear the authors' interpretation of the results (and I believe that the reader would be happy to see them as well).

Correctness: The proofs in the paper are easy to follow and rigorous.

Clarity: The paper is very well written and well-organized.

Relation to Prior Work: Previous work is well discussed. The game-theoretic literature on dynamic equilibrium for repeated games should also be mentioned. For many people, a repeated game is more than a learning scenario with T rounds.

Reproducibility: Yes

Additional Feedback: The paper indeed generalizes repeated games, but the equilibrium analysis definitely doesn't generalize the equilibria of repeated games (i.e., subprefect equilibria and Folk Theorem). In other words, the policy here is limited to be a (random) function of the context, and not the strategy profile (which shouldn't be confused with the fact that this function itself is indeed a function of the history of the game). To avoid confusion, it's better to discuss this issue. In light of this, statements like in lines 18-19 are a bit misleading and maybe even lines 102-103 (which are technically correct though). The structure in 90-92 should include the fact that players get to observe the strategy profile. Line 129 is much too late for that. That's a strong and crucial assumption. Line 98 is a bit misleading. If a player would know the actions of other players ahead of time, it would change its actions which will change their actions in response (even with your algorithm that doesn't respond to the strategy profile). I think the precise statement is more technical/dry than that. Please write GP (Gaussian Process) explicitly. I think it's better to state the Theorem for Section 3.1 (which is currently in the appendix). The assumption in Line 183 should be mentioned earlier, perhaps together with the regularity assumptions. The connection between this assumption and the regularity assumption might be worth discussing. The L1 in Line 192 should be in math font. Line 199 (and Line 239) sneaks in that the number of rounds T is known. That's not a trivial assumption to make since I don't see how a doubling trick can be applied here (a discussion on this issue can be interesting). In fact, it's not clearly mentioned that the game is played for T rounds. For example, it's not like Proposition 4 holds for any T, and the approximation gets better with time in an online manner. It only holds if the parameters were tuned using T. Please clarify and explicitly states these facts (if correct). It's better to state how \gamma_T can be bounded for different kernels below the statement of Theorem 1, otherwise, it's not clear if this term is significant or not (which line 169 does make clear). This is quite subjective, but I didn't really understand the motivation for the i.i.d. contexts case. I think it diverges from the main point of the paper so you might want to consider moving it to the appendix (moving in Theorem 9, or even Lemma 7 and 8 that are highly relevant to the understanding of your algorithm). Specifically: 1) I don't see why the assumptions in Lines 216-217 are natural or constitute an interesting case. Eventually, since players observe the contexts, they would be able to approximate its distribution and sample from it. Also, how can the context be a private type of a specific player, if it's the same context for everyone (a vector of types?) that then gets revealed to everyone? 2) Why wouldn't the opponent care about the type of the player? taking it into account can improve its outcome. 3) Isn't this case equivalent to adding another player, making z_t his mixed strategy (so i.i.d. overtime), and just studying the correlated equilibrium of this N+1-player game? The definition of the contextual-coarse-correlated equilibria, while being very clear, is also somewhat more technical than meaningful. As opposed to correlated equilibrium, which depends on the static game, it's hard to imagine why would one want to converge or to compute this equilibrium which is very empirical by nature (it's an equilibrium with respect to the context sequence that was observed in the T turns that constitute the game, where T is known). While I understand that for an arbitrary context sequence there isn't much better than can be done, this is a bit disappointing as an equilibrium notion and sort of implies that an arbitrary sequence of contexts is so strong it prevents a more compact equilibrium notion. Definition 10 for the i.i.d. context case, which is more compact supports this impression. I think that the conclusions are a good place to discuss limitations and ideas for future research. This is especially true for a paper that provides a new nice framework that people might be interested in. I feel a bit uncomfortable that the proof for Lemma 7 appears in a dissertation that's not peer-reviewed. Can you please provide another reference? I think the application of Hoeffding–Azuma inequality in (13) deserves a more careful explanation. The issue is that all the randomness here is conditioned on the event that all the estimations fall within the confidence intervals. Then arguing that this is still a martingale-difference sequence can become subtle. Having said that, I'm not sure why (13) only holds with high probability. It looks like only the fact that this sum represents something of interest is true with high probability. Please clarify. ***************************************************************************************************************************************************************** Post Rebuttal: Thank you for your response, It clarifies a misconception I had about the motivation of the paper, so I'm training my score from 6 to 7. This misconception has to do with how much this paper is about adversarial contextual bandits versus how much it is about games (as the title suggests). It is clearly not just an adversarial contextual bandit, as lines 133-141 explain well. The additional assumption that the "adversary" is in fact N-1 other players that player i can observe their actions allows for stronger results. However, and that's subjective of course, the paper has less to offer on the game-theoretic side, for example regarding the dynamics or the equilibrium. I think the writing can be clearer and sharper about this motivation. Specifically: 1) The strong results are Theorem 1 and Theorem 2, which argue about the perspective of some single player i, even if other players don't run your algorithm. Although it is stated, to avoid confusion, it's better to emphasize that these results don't assume in any way that all players run your algorithm (so when you write "player i" it does not necessarily hold for each i). On the same note, a_{t}^{-i} in Theorem 2 should be explicitly defined to avoid any wrong ideas from the reader about how a_{t}^{-i} is generated. 2) The results of Section 4 on "Game Equilibria and Efficiency", in my opinion, still suffer from the main weakness I mentioned above. It's unlikely that in this setting, where players can observe the past actions of all others, *all* players will run your algorithm, and none of them will be tempted to manipulate the others in more sophisticated ways that this dynamic setting allows (i.e., by responding to the strategy profile, which your algorithm doesn't directly do). Hence, while I agree that due to the robustness and performance guarantees of your algorithm, it makes sense that *some* players will run it, the results of this section hold, as you properly state, only if all of them do. However, this can indeed be the case for cooperative players, which is why I believe it's an interesting direction to elaborate on (or for future research). 3) With single-player as the main motivation, Section 3.3 makes more sense. To avoid confusion, it's better to emphasize that the idea is that other players don't run your algorithm, and the context indeed concerns only player i. Lines 215-220 are a good start, but it needs to be clear that "player i" is not a general way to refer to all of the players. 4) The regret is a random variable. Some of your statements use its realization (Proposition 4 and 11), which is fine. But in some others, the meaning is vaguer. In lines 110, line 587, and line 601 (Proposition 11), I guess you mean that R(T)/T needs to vanish with probability 1. Please clarify and explain why the result of Theorem 1 indeed implies that (Borel-Cantelli?).


Review 3

Summary and Contributions: The paper considers a repeated game setting, where games may be different in each round, but are characterized by a round-dependent known context. The context impacts the game pay-offs, and provided the difference in contexts bounds the difference in pay-offs, the authors present algorithms to play contextual games with vanishing (contextual) regret and relate these results to contextual bandits. They then study the equilibrium outcome and show that players converge to this equilibrium when they play no-regret algorithms (which mirrors the result for the static game case). The authors also show that the welfare of the equilibrium can be bounded for smooth games. Finally they complement their theoretical work with an empirical analysis of contextual traffic routing.

Strengths: 1. The model is well-motivated and supported with empirical evaluation. The connection to contextual bandits is also well-received. 2. I appreciated the definition and analysis of equilibrium that arises as a consequence of these repeated contextual games.

Weaknesses: 1. It would have been nice to study the equilibrium quality empirically as well, in addition to the evaluation of the algorithms.

Correctness: The claims and proofs seem correct.

Clarity: The paper is clear.

Relation to Prior Work: yes

Reproducibility: Yes

Additional Feedback: I was wondering how the notion of contextual game relates to that of succinct games (which are described by a polynomial number of parameters, see “Computing Correlated Equilibria in Multi-Player Games”). POST-REBUTTAL I appreciate the authors responses. For welfare (I didn't do a good job describing this), I'd be interested to understand the quality of the equilibrium compared to optimal welfare (say, an empirical version of price of anarchy). The authors introduce a new equilibrium concept (which is quite broad by analogy of CCE being a broad equilibrium class), but there's no discussion of whether solutions to this equilibrium concept have reasonable welfare (compared to optimal). What the authors point out is that the plots show that the welfare comparisons against benchmarks that perform worse, but they don't show it against the optimal benchmark. I won't hold that interpretation of my comment against the authors, and I've kept the score the same.


Review 4

Summary and Contributions: This paper introduces a class of games called contextual games. This is a repeated game where the players receive an observable “context” at each round. Assumption leveraged by the rest of the paper is that similar contexts lead to similar outcomes (formalized through kernel-based regularity). The authors also introduce contextual regret and algorithms for minimizing this notion. Analogically, contextual (coarse) correlated equilibria are also introduced as they are closely related to minimizing the contextual regret. The paper also includes an experiment on a traffic routing problem, where the context semantics is capacity of the network at each round.

Strengths: I enjoyed the paper, it is well motivated and the introduced concepts are clear and well defined. The important results are also theoretically grounded, and the presented theorems make sense. The introduced experiment also does not feel artificial, and shows that the introduced methods and context can be practically used.

Weaknesses: I feel like a limitation is the feedback model. Authors assume that after each round, not only the agent receives (noisy) observation, but also observes the actions of all the other players. This assumption is in many settings unrealistic. While I am not an expert in this particular line of work,some of the results also look like relatively simple combinations of previous work. This is not necessarily bad, as the authors do a very good job of referring to the previous results as they introduce their techniques (which I appreciate). It still sometimes feels like just putting existing things together in a straightforward way (e.g. GP-MW -> C.GP-MW, Strategy 2) Minor limitation is that the introduced models and techniques do not deal with sequential settings, but that could arguably be extended (maybe as a future work)

Correctness: All seems correct.

Clarity: The paper is clearly written and reads well.

Relation to Prior Work: Previous work seems to be well referenced.

Reproducibility: Yes

Additional Feedback: Line 183: I think you are missing z_t as a parameter for the optimal policy function Experiment: Maybe add a note that your reward function satisfies your assumptions. ***************************************************************************************************************************************************************** Post Rebuttal: No change to my score, I liked the paper and was happy with the feedback provided to the other reviewer's comments.

[Author Response · NeurIPS 2020]

We would like to thank all the reviewers for their constructive feedback. In the following, we respond (**R**) to individual concerns (**C**) summarized in italic. Citations refer to references in the paper.

**Reviewer 1. C:** *"...how does the term $\gamma_T$ change as we increase number of actions $K$, and number of agents $N$?"* **R:** The Reviewer is correct in that the term $\gamma_T$ can dominate the regret bound, however for most widely used kernels $\gamma_T$ grows only with the dimension $d$ of the domain $\mathcal{D} = \Pi_{i=1}^{N} \mathcal{A}^i \times \mathcal{Z}$ and *not* with the number of actions $K = |\mathcal{A}^i|$ available to player $i$. Note that although the possible action tuples grow exponentially with the number of players $N$, the dimension $d$ grows only linearly with $N$. **C:** *"...Why should the price of anarchy be bad if there is just one round t that gives poor choices of $\lambda(z_t)$ and $\mu(z_t)$?"* **R:** The obtained price of anarchy bound reflects the worst case in which, if the game is non-smooth even for a single round, the welfare function $\Gamma$ for that round could have a much higher contribution than the others and thus highly deteriorate the performance. Nevertheless we agree that, based on the Reviewer's reasoning, tighter conditions could perhaps be found as a function of the contexts' sequence. Finally, we will clarify the connections with respect to the mentioned previous works [Slivkins, 2011] and [Lu et al. 2010] that study similarity in contextual bandits. Lu et al. consider a stochastic setting and exploits Lipschitzness of the single reward function $r(a_t, z_t)$, while Slivkins studies similarity even in contextual adversarial bandits (i.e., where the player faces an adversarial sequence of reward functions $r_t(a_t, z_t)$). In a contextual game the rewards obtained by player $i$ are generated by the game reward function $r(\cdot)$ which, however, also depends on the actions $a_t^{-i}$ chosen by the other players. That is, a contextual game is an adversarial contextual bandit problem where $r_t(a_t, z_t) = r(a_t, a_t^{-i}, z_t)$. This makes our model unique and different from previous work in that we impose kernel-based regularity assumption on $r(\cdot)$ and study the similarity across different game outcomes $(a_t, a_t^{-i}, z_t)$. We will clarify these aspects in the paper.

**Reviewer 2. C:** *"...it's not clear why would players agree to run Algorithm 1 while they can adopt dynamic sophisticated strategies (i.e., Folk Theorems)"* **R:** Although players are aware of playing against N-1 agents, these agents may be non-rational or even adversarial. In such a case we believe the proposed no-regret algorithms represent a simpler and more robust playing choice for the agents, which come with individual performance guarantees, as opposed to using Folk theorems. Moreover, they do not require the game to be known in advance. This certainty leads to a simpler non-dynamic equilibrium notion (c-CCE), which however can certify a certain level of game welfare as discussed in Section 4.
**C:** *"The paper indeed generalizes repeated games, but the equilibrium analysis definitely doesn't generalize the equilibria of repeated games"* **R:** We agree with the Reviewer, and we will make this clear. Our notion is a simpler, non-dynamic, and more tractable notion of equilibrium which takes into account the context information but does not have 'memory' of past rounds. The repeated game is essentially summarized with its time-average and, in this regard, c-CCEs generalize CCEs for non-contextual games (where the time-average coincides with the one-shot game).
**C:** *"...it's not like Proposition 4 holds for any T, and the approximation gets better with time in an online manner. It only holds if the parameters were tuned using T."* **R:** We remark that the horizon $T$ does not necessarily need to be known in advance. In Theorem 1 the doubling trick can be used to tune the radius $\epsilon$ (see, e.g., [20, Remark 1]), while in Theorem 2 a time-varying learning rate $\eta$ can be chosen (see, e.g., [11, Theorem 2.3]). Hence, Proposition 4 holds for any $T$ and the c-CCE approximation gets better with time in an online manner. We will clarify this in the paper.
**C:** *"This is quite subjective, but I didn't really understand the motivation for the i.i.d. contexts case."* **R:** We agree with the Reviewer that the discussion in Section 3.3 diverges from the main point of the paper; we decided to include it only as a special case in which players can achieve significantly improved performance. However, note that even if players perfectly knew the contexts distribution, the pseudo-regret in Theorem 2 is a function of the *realized* contexts and therefore identifies a non trivial benchmark. Moreover, our motivation for Section 3.3 are settings in which the context is private information only relevant to player $i$ (e.g., value for items in an auction, or production costs in a market) and therefore the other players do not have access to it (they can still base their decision on other private contextual information). Finally, we thank the Reviewer for other comments that can improve the exposition of the paper.

**Reviewer 3. C:** *"It would have been nice to study the equilibrium quality empirically as well, in addition to the evaluation of the algorithms."* **R:** Besides comparing the players' individual performance, Figure 1 also shows that the game welfare (i.e., the sum of players' payoffs) increases and the network congestion level decreases as the players approach a contextual CCE. Hence, it empirically demonstrates the quality of the computed c-CCE with respect to playing non-contextual CCEs (cyan line) and not learning (purple line).
**C:** *"I was wondering how the notion of contextual game relates to that of succinct games."* **R:** We believe the notion of a game having a succinct representation can co-exist with our contextual game setup, as a function of the contextual information. For instance, for any given context (e.g., network occupancy profile) the considered traffic routing game is a congestion game (hence a particular kind of succinct game). It is an interesting future direction to understand if the computational advantages of finding correlated equilibria in succinct games can be lifted to compute contextual correlated equilibria.

**Reviewer 4.** We thank the Reviewer for the positive feedback and comments. It is indeed an interesting future work to extend the present models and techniques to sequential settings.

[Meta-Review · NeurIPS 2020]

The reviewers agree that this is a good contribution to the literature on learning in games. The authors are strongly encouraged to improve presentation regarding how the various constants (e.g. gamma_T) depend on primitive quantities, in different settings.